# ST-SimDiff: Balancing Spatiotemporal Similarity and Difference for Efficient Video Understanding with MLLMs

**Bingjun Luo**[1], **Tony Wang**[1], **Chaoqi Chen**[2]*, **Xinpeng Ding**[3]

[1]Tsinghua University
[2]Shenzhen University
[3]Xidian University
bingjunluo@outlook.com, tonywang5454@gmail.com,
cqchen1994@gmail.com, xdingaf@connect.ust.hk

## Abstract

Multimodal Large Language Models (MLLMs) face significant computational overhead when processing long videos due to the massive number of visual tokens required. To improve efficiency, existing methods primarily reduce redundancy by pruning or merging tokens based on importance or similarity. However, these approaches largely overlook a critical dimension of video content, i.e., changes and turning points, and they lack a collaborative model for spatio-temporal relationships. To address this, we propose a new perspective: similarity is for identifying redundancy, while difference is for capturing key events. Based on this, we designed a training-free framework named ST-SimDiff. We first construct a spatio-temporal graph from the visual tokens to uniformly model their complex associations. Subsequently, we employ a parallel dual-selection strategy: 1) similarity-based selection uses community detection to retain representative tokens, compressing static information; 2) temporal difference-based selection precisely locates content-changing points to preserve tokens that capture key dynamic shifts. This allows it to preserve both static and dynamic content with a minimal number of tokens. Extensive experiments show our method significantly outperforms state-of-the-art approaches while substantially reducing computational costs. Our code is available in https://github.com/bingjunluo/ST-SimDiff .

## 1 Introduction

The rise of Large Language Models (LLMs) has significantly propelled advancements of Large Vision-Language Models (LVLMs), which have demonstrated remarkable capabilities in image and video understanding (Zhang et al., 2024c; Liu et al., 2024c). For video processing, current LVLMs typically sample a video into a sequence of frames and then convert each frame into hundreds or even thousands of visual tokens for the LLM (Bai et al., 2025; Li et al., 2024). While this paradigm is effective, the number of tokens grows explosively with increasing video duration and resolution. This leads to prohibitive computational and storage burdens, severely limiting the application of LVLMs in scenarios such as long-video analysis and real-time interaction (Fu et al., 2024b).

To address this challenge, various methods are proposed to enhance the efficiency of LVLMs. These approaches can be categorized into two types. One category is importance-based pruning. As observed by FastV (Chen et al., 2024), there is redundancy in the attention distribution across different layers when LVLMs process visual information. Therefore, visual tokens with lower attention scores in the deeper layers of the model can be pruned to reduce the computational load. The other category is similarity-based merging/selection. Existing works have found high similarity among visual tokens, either between adjacent video frames or within a single frame. FrameFusion (Fu et al., 2024b) reduces redundancy by merging similar tokens from adjacent frames, while VisionZip (Yang et al., 2025) directly selects dominant tokens at the vision encoder level.

---

*Corresponding author.

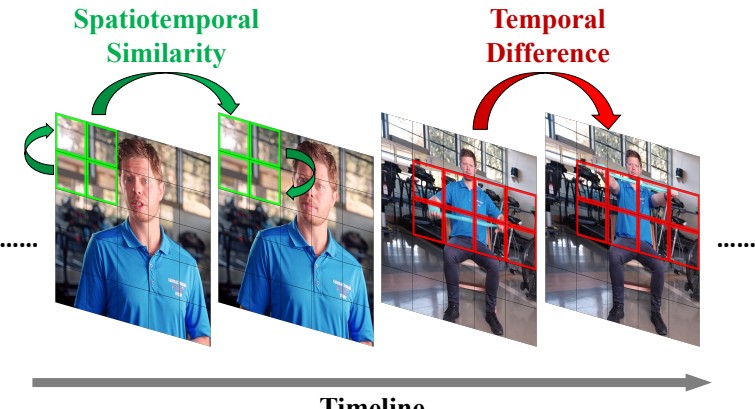

Figure 1: The core motivation of ST-SimDiff. We posit that efficient video understanding requires handling two scenarios simultaneously: Spatiotemporal Similarity (left) can be utilized to compress redundant information both spatially within a frame and temporally across adjacent frames. Temporal Difference (right) should be detected to capture the key actions or events that define the plot.

Despite the success of existing methods in visual token reduction, they face two key limitations. First, existing token pruning methods often focus on spatial correlations within the same frame or temporal correlations at the same positions across different frames. This lack of joint modeling and analysis of spatio-temporal correlations prevents them from effectively capturing complex dynamic events. Furthermore, these methods share a potential blind spot: they mainly focus on informational commonalities, like similarity and importance, while neglecting crucial changes and differences. Since the narrative of the video is often driven by turning events, a model that only seeks similarity might smooth over a sudden action, leading to a misinterpretation of the content.

Based on this, we propose a new perspective: similarity is for identifying redundancy, while difference is for capturing key moments. We believe that an ideal video token compression algorithm should achieve two goals simultaneously: representing the stable content of a video with the fewest tokens, and precisely preserving the key changes. To this end, we propose **ST-SimDiff**, a dual token selection framework based on a spatio-temporal graph. Specifically, the framework consists of two parts. The first is **similarity-based representative token selection**: we treat all visual tokens of a video as nodes and construct a graph based on their feature similarity in the spatio-temporal dimension. Using a graph community detection algorithm, we find tightly connected token clusters in the graph, which represent stable and persistent visual elements in the video. We select a few central tokens with higher centrality from each cluster as representatives. The second part is **difference-based event token selection**: we pay special attention to the edges connecting along the time axis in the graph. When the similarity between corresponding tokens of adjacent frames drops sharply, we consider it a turning point and mark these tokens as critical event tokens that must be retained. Finally, we merge these two sets of tokens to be input into the LLM. Our contributions can be summarized as follows:

- We are the first to propose that the similarity and difference of tokens should be given equal importance in VLM efficiency research.

- We design a spatio-temporal graph framework to uniformly model the complex spatio-temporal relationships between video tokens. We propose a novel dual token selection strategy that can simultaneously screen for both representative and pivotal event tokens.

- We conduct extensive experiments on video understanding benchmarks, and the results show that ST-SimDiff significantly reduces the visual token budget while maintaining or even improving performance.

## 2 RELATED WORK

### 2.1 LARGE VISION LANGUAGE MODELS

The rapid advancement of Large Language Models (LLMs) (Touvron et al., 2023; Achiam et al., 2023; Ouyang et al., 2022) has significantly propelled progress in multimodal understanding, leading to the emergence of numerous prominent Large Vision Language Models (LVLMs) (Liu et al., 2024b; Bai et al., 2023). These LVLMs typically process images or video frames by converting them into visual tokens via pre-trained visual encoders, which are then fed into LLMs. Various alignment modules commonly connect the visual encoder and the LLM, including MLPs (Liu et al., 2024a;b) and Q-Formers (Li et al., 2023; Dai et al., 2023). This architecture enables LVLMs to exhibit exceptional capabilities in multimodal tasks (Xi et al., 2025; Holdaway et al., 2024; Zhang et al., 2025a). Particularly in video understanding, they show promise in processing longer, higher-resolution complex videos (Bai et al., 2025; Li et al., 2024; Zhang et al., 2024c; Liu et al., 2024c).

However, this powerful capability also introduces significant computational challenges, especially in video understanding scenarios. Due to the need to process continuous sequences of frames, the number of visual tokens grows exponentially, far exceeding that of static images, posing a severe challenge to the training and inference efficiency of LVLMs. Therefore, how to efficiently process massive video visual tokens and reduce computational overhead remains a critical issue in current LVLM research.

### 2.2 VISUAL TOKEN COMPRESSION

Driven by the substantial overhead of video processing, visual token compression has become a critical method for improving the efficiency of Large Visual Language Models (LVLMs). Existing methods primarily approach compression from two aspects: token importance and similarity.

Earlier methods primarily focused on importance-based token pruning, aiming to identify and remove visual tokens that contribute less to model performance. These methods typically utilize attention scores or other feature metrics to quantify token importance, such as FastV (Chen et al., 2024) and FasterVLM (Zhang et al., 2024b). Subsequent works (Shang et al., 2024; Ye et al., 2025) have started to further consider token similarity to more effectively reduce redundancy and preserve critical information. These methods recognize that even important tokens can exhibit high similarity, leading to informational redundancy. For instance, FrameFusion (Fu et al., 2024b) combines similarity-based merging with importance-based pruning. VisionZip (Yang et al., 2025) reduces redundancy by selecting dominant tokens and merging contextual information. VisPruner (Zhang et al., 2025b) leverages visual cues to remove redundant items.

Despite significant progress in visual token redundancy, existing work hardly addresses the intricate, global spatio-temporal similarity relationships between tokens. In video scenarios, visual information is redundant both spatially and temporally. Existing similarity-driven methods, like FrameFusion, mainly only on the temporal similarity between adjacent frames, failing to utilize complex spatio-temporal relevant information. This limits their ability to fully exploit inherent video redundancy, restricting performance in extreme token reduction scenarios.

## 3 PROBLEM DEFINITION

Given a video $V$ and a text query $Q$, a standard Large Vision-Language Model (LVLM) first encodes the video into a full sequence of $N$ visual tokens, $T_{\text{full}} = \{t_1, t_2, \ldots, t_N\}$. This process is computationally demanding, as the self-attention mechanism's complexity scales quadratically with the token count $N$, i.e., $O(N^2)$. Our task is therefore to design an efficient token selection function, $f(\cdot)$, that takes the full sequence $T_{\text{full}}$ and a compression ratio $r$ to produce a much smaller subset $T_{\text{sub}} = f(T_{\text{full}}, r)$. The objective is to maximize the LVLM's downstream task performance using this subset, subject to the constraint that its size $|T_{\text{sub}}|$ is approximately $r \cdot N$. The core challenge is to design the selection function $f(\cdot)$ to retain the most critical information within the subset $T_{\text{sub}}$.

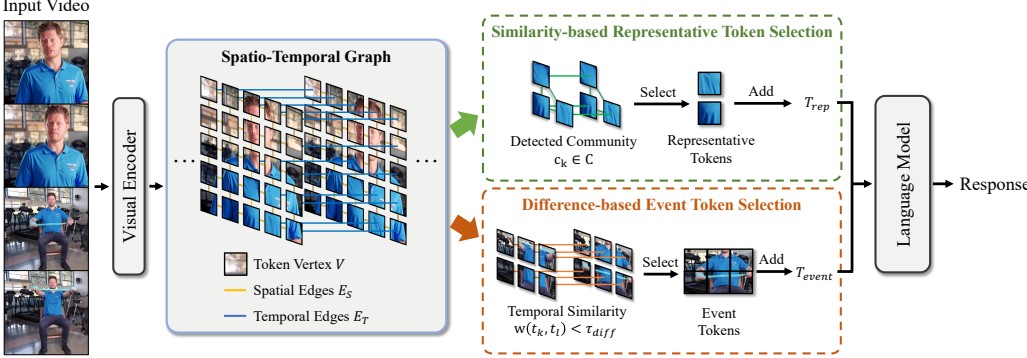

Figure 2: The overview framework of ST-SimDiff, which consists of three parts: Spatio-Temporal Graph Construction, Similarity-based Representative Token Selection, and Difference-based Event Token Selection.

# 4 ST-SIMDIFF

## 4.1 METHOD OVERVIEW

As illustrated in Figure 2, the proposed ST-SimDiff method comprises two key components that operate in parallel on a spatio-temporal graph constructed from the video's visual tokens: Similarity-based Representative Token Selection (SRTS) and Difference-based Event Token Selection (DETS). First, in the SRTS module, we identify and condense the video's stable and redundant content. By applying community detection algorithms to the graph, we group highly similar tokens into clusters, which correspond to persistent visual elements like static backgrounds. From each cluster, we then select a few highly central tokens to form a representative set, $T_{\text{rep}}$. Second, the DETS module is designed to capture the video's crucial dynamic shifts. It analyzes the temporal edges of the graph, identifying moments where the similarity between corresponding tokens in adjacent frames drops sharply. These points of high difference signify turning points, and the tokens framing these transitions are preserved as critical event tokens, forming the set $T_{\text{trans}}$. By synergistically combining these two components, ST-SimDiff generates a final token subset $T_{\text{sub}} = T_{\text{rep}} \bigcup T_{\text{trans}}$, which is both highly compact and information-rich, retaining stable content while precisely capturing key events.

## 4.2 SPATIO-TEMPORAL GRAPH CONSTRUCTION

Given a video, the vision encoder transforms it into a sequence of tokens $T = \{t_1, t_2, \ldots, t_N\}$. Each token $t_k \in T$ corresponds to a specific spatio-temporal location and is associated with a feature vector $\mathbf{x}_k$. We define the location of each token $t_k$ by its frame index $T(t_k)$, its spatial height index $H(t_k)$, and its spatial width index $W(t_k)$.

We model the relationships between these tokens using a spatio-temporal graph $G = (V, E)$, where the vertex set $V$ represents all tokens in $T$. The edge set $E$ is the union of two distinct subsets, $E = E_S \cup E_T$, which capture spatial and temporal relationships respectively. The spatial edges $E_S$ connects tokens that are spatially adjacent within the same frame:

$$E_S = \{(v_i, v_j) \in V \times V \mid T(t_i) = T(t_j) \text{ and} \\ |H(t_i) - H(t_j)| + |W(t_i) - W(t_j)| = 1\} \tag{1}$$

The temporal edges $E_T$ connects tokens that are at the same spatial position but in adjacent frames, capturing temporal continuity:

$$E_T = \{(v_i, v_j) \in V \times V \mid H(t_i) = H(t_j), \\ W(t_i) = W(t_j), \text{ and } |T(t_i) - T(t_j)| = 1\} \tag{2}$$

The weight of any edge $w(v_i, v_j) \in E$ is defined by the cosine similarity of the feature vectors of the corresponding tokens:

$$w(v_i, v_j) = \frac{\mathbf{x}_i \cdot \mathbf{x}_j}{\|\mathbf{x}_i\|\|\mathbf{x}_j\|} \tag{3}$$

This sparse graph structure efficiently encodes local spatial relationships and temporal continuity, providing a unified foundation for our dual-path token selection strategy.

### 4.3 Similarity-based Representative Token Selection

Video content is inherently characterized by substantial spatio-temporal redundancy. For instance, in a static scene that persists for several seconds, the background tokens exhibit high similarity over time. Similarly, for an object moving across the screen, its corresponding tokens, despite changing spatial positions, maintain a high degree of semantic correlation. Existing methods often address temporal and spatial similarities in isolation. In contrast, our spatio-temporal graph is designed to capture these joint similarities. Within this graph, semantically related tokens form dense "communities" or "clusters" based on their feature similarity, irrespective of their specific frame or location. Therefore, selecting representatives from each community emerges as a highly efficient strategy for compressing redundant information while preserving core semantics.

To accurately identify strongly correlated token clusters, we first threshold the graph $G$. We set a similarity threshold $\tau_{\text{sim}}$ and retain only the edges with weights above this threshold, forming a new graph $G' = (V, E')$, where $E' = \{(v_i, v_j) \in E \mid w(v_i, v_j) > \tau_{\text{sim}}\}$. Next, we apply a graph community detection algorithm (e.g., the Louvain method (Blondel et al., 2008)) on $G'$ to identify token clusters $C = \{c_1, c_2, \ldots, c_m\}$. For each community $c_k \in C$, we rank and filter its internal tokens based on their centrality. The centrality score $S_c(t_a)$ of a token $t_a \in c_k$ is defined as its average similarity with all other tokens within the community:

$$S_c(t_a) = \frac{1}{|c_k| - 1} \sum_{t_b \in c_k, b \neq a} w(t_a, t_b) \tag{4}$$

Subsequently, we set the intra-community retention rate to the globally preset compression ratio $r$, and preserve the top $\lceil |c_k| \cdot r \rceil$ tokens with the highest centrality scores from each community $c_k$. This uniform filtering process naturally handles all communities, including single-node communities composed of unique tokens (in which case the sole node is retained). Formally, the set of representative tokens, $T_{\text{rep}}$, is defined as:

$$T_{\text{rep}} = \bigcup_{k=1}^{m} \underset{t \in c_k, \text{score}=S_c(t)}{\text{TopK}} (\lceil |c_k| \cdot r \rceil) \tag{5}$$

where the TopK function returns the set of $k$ elements with the highest scores from a given set. In summary, this strategy efficiently compresses the static and persistent content of the video by identifying and retaining the most central tokens within each semantic cluster.

### 4.4 Difference-based Event Token Selection

If similarity defines the "norm" of a video, then difference defines its "events". A model focusing only on similarity may excel at understanding "what is" but struggle with "what happened". A video's plot is driven by key events, and the essence of an event is change—the appearance of a new object, the start of an action, or a scene transition. In our model, these changes manifest as a sharp difference in the features of temporally adjacent tokens. Therefore, accurately capturing these "turning points" of significant difference is crucial for understanding the video's dynamic content and correctly answering questions like "when" and "why".

Difference, particularly along the temporal dimension, often signals the occurrence of a key event. For example, the entry of a new object, a scene change, or the beginning/end of an action will cause drastic changes in the visual features at corresponding positions in adjacent frames. We specifically analyze the temporal edges ($E_T$) in our spatio–temporal graph. For any temporal edge $(v_k, v_l) \in E_T$ in the graph, which connects two temporally adjacent tokens $t_k$ and $t_l$, we set a dynamic threshold $\tau$ (e.g., the 95th percentile of all temporal edge difference scores). When the difference score of a temporal edge $(v_k, v_l)$ exceeds this threshold, i.e., $w(t_k, t_l) < \tau_{\text{diff}}$, we consider the subsequent token $t_l$ as a critical event token and retain it. Formally, the set of event tokens, $T_{\text{event}}$, is defined as:

$$\begin{aligned} T_{\text{event}} = \{ t_l \mid &\exists t_k \text{ s.t. } (v_k, v_l) \in E_T, \\ &T(t_l) > T(t_k), \text{ and } w(v_k, v_l) < \tau_{\text{diff}} \} \end{aligned} \tag{6}$$

With this strategy, tokens that signify moments of significant temporal change can be preserved. By capturing these key transitions, we ensure that the crucial dynamic aspects of the video's narrative are not overlooked during the compression process.

## 4.5 OVERALL REDUCTION PROCESS

In the proposed framework, we first compute the representative token set $T_{\text{rep}}$ and the event token set $T_{\text{event}}$ in parallel, and take their union, $T_{\text{candidate}} = T_{\text{rep}} \cup T_{\text{event}}$, as the initial candidate set. To precisely meet the target token count $N_{\text{target}} = \lceil r \cdot N \rceil$, we introduce a final pruning step. If the size of the candidate set, $|T_{\text{candidate}}|$, exceeds $N_{\text{target}}$, we remove the $|T_{\text{candidate}}| - N_{\text{target}}$ tokens with the lowest importance. Following existing work (Chen et al., 2024), the importance of a token is determined by its attention score in the shallow layers of the LLM. This step ensures that our method can flexibly meet any computational budget while preserving critical information.

In summary, our overall method adeptly combines two strategies: graph-based token selection and attention-based dynamic pruning. The former leverages the intrinsic structure of the video content (similarity and difference) to ensure the retention of core information, while the latter provides a flexible mechanism to precisely meet any given computational budget. This design makes our token compression process both principled and adaptive, thereby achieving a robust balance between efficiency and performance.

## 4.6 COMPUTATIONAL COMPLEXITY ANALYSIS

The computational overhead of our proposed ST-SimDiff framework can be analyzed in three main stages: Spatio-Temporal Graph Construction, Similarity-based Representative Token Selection (SRTS), and Difference-based Event Token Selection (DETS).

The initial stage involves building the graph. For each of the $N$ visual tokens, we compute the cosine similarity with its spatially and temporally adjacent neighbors. As each token has a small, constant number of neighbors, this step requires a single pass over all tokens, resulting in a complexity of $O(Nd)$, where $d$ is the feature dimension of each token.

Following graph construction, the SRTS module identifies token clusters. As detailed in our implementation (Section 5.2), we employ a connected components algorithm, which operates with a near-linear time complexity of $O(N + |E|)$. Since the number of edges $|E|$ is at most $3N$, this simplifies to $O(N)$. The subsequent filtering step within each community could be computationally intensive. A naive approach of calculating all-pairs similarity within each community $c_k$ would have a complexity of $O(|C| \cdot |c_k|^2 \cdot d)$. To prevent this, we impose a constraint on the maximum size of any community. In the rare event that a community exceeds a predefined threshold (e.g., $|c_k| > \sqrt{N}$), we partition it. This ensures the filtering process remains efficient and does not dominate the overall complexity. While other mainstream graph clustering algorithms such as the Louvain and Leiden methods offer more complex community definitions, their computational complexity is higher, approaching $O(N \log N)$. Therefore, we opted for the simpler and highly efficient connected components algorithm to ensure minimal processing overhead.

The DETS module involves a single pass through all temporal edges in the graph to identify significant changes by comparing token similarities against a threshold. This process has a linear complexity of $O(Nd)$.

In summary, the total complexity of our ST-SimDiff framework is governed by these linear-time operations, culminating in an overall complexity of $O(Nd)$. This is substantially more efficient than the quadratic complexity of the self-attention mechanism, $O(N^2 d)$, used in the Large Language Model. Therefore, our method provides significant token reduction with only a negligible computational footprint relative to the model's main inference cost.

| Method | VideoMME | | | | LongVideoBench | | | EgoSchema |
|---|---|---|---|---|---|---|---|---|
| | Short | Medium | Long | Overall | Relation | Perception | Overall | |
| **Upper Bound (Full Performance)** | | | | | | | | |
| LLaVA-Video | - | - | - | 63.3 | - | - | 58.2 | 57.3 |
| **Token Retain Ratio $r = 30\%$** | | | | | | | | |
| FastV | 68.2 | 58.6 | 51.2 | 59.3 | 48.0 | 59.9 | 53.5 | 51.3 |
| PruMerge | 70.2 | 57.3 | 52.2 | 59.9 | 49.6 | 60.5 | 54.7 | 50.9 |
| FasterVLM | 71.3 | 57.6 | 51.4 | 60.1 | 51.0 | 61.3 | 55.8 | 52.6 |
| VisionZip | 68.4 | 55.9 | 50.4 | 58.3 | 46.9 | 60.3 | 53.2 | 53.0 |
| FrameFusion | 74.0 | 59.8 | 50.0 | 61.3 | 49.7 | **63.3** | 56.0 | 53.0 |
| **ST-SimDiff (Ours)** | **74.7** | **61.9** | **53.0** | **63.2** | **52.5** | 63.2 | **57.5** | **56.0** |
| **Token Retain Ratio $r = 50\%$** | | | | | | | | |
| FastV | 73.9 | 61.4 | 51.3 | 62.2 | 50.1 | 62.2 | 55.7 | 54.7 |
| PruMerge | 72.3 | 58.7 | 52.8 | 61.3 | 51.4 | 63.2 | 56.9 | 54.6 |
| FasterVLM | 74.0 | 59.2 | 51.8 | 61.7 | 51.3 | 62.2 | 56.4 | 56.2 |
| VisionZip | 72.6 | 57.7 | 51.6 | 60.6 | 51.3 | 63.2 | 56.8 | 54.2 |
| FrameFusion | 74.6 | 61.2 | 52.0 | 62.6 | 51.7 | 64.2 | 57.6 | 55.8 |
| **ST-SimDiff (Ours)** | **76.2** | **62.4** | **52.7** | **63.8** | **52.3** | **64.3** | **57.9** | **57.3** |

Table 1: Performance comparison on long-form video understanding benchmarks on LLaVA-Video-7B for 64 input frames (%). The best performance among all methods is emphasized in **bold**.

## 5 EXPERIMENTS

### 5.1 EXPERIMENTAL SETUP

**Baselines** To comprehensively evaluate our method's effectiveness, we use LLaVA-Video (Zhang et al., 2024c) and NVILA (Liu et al., 2024c) as our base models and compare our approach against a range of state-of-the-art video token compression techniques. These baselines cover diverse strategies, including **importance-based reduction methods** like FastV (Chen et al., 2024) and Faster-VLM (Zhang et al., 2024b), and **hybrid reduction methods** like VisionZip (Yang et al., 2025), FrameFusion (Fu et al., 2024b), and PruMerge (Shang et al., 2024). To ensure a fair and complete comparison, we also include the original uncompressed model (Vanilla) as a performance upper bound and Random Sampling of an equivalent number of tokens as a lower bound.

**Benchmarks** To test our method's capabilities in long-video understanding, we adopt three challenging benchmark datasets: VideoMME (Fu et al., 2024a), LongVideoBench (Wu et al., 2025), and EgoSchema (Mangalam et al., 2024). VideoMME is a comprehensive benchmark for foundational video understanding, featuring diverse video types and three length categories: Short, Medium, and Long. LongVideoBench evaluates the ability to retrieve and reason about details in long videos through a "referring reasoning" task, with videos of four lengths: 15s, 60s, 600s, and 3600s. EgoSchema is a question-answering dataset focused on long-term, first-person videos that tests the model's ability to understand character intentions and causal chains of actions.

### 5.2 IMPLEMENTATION DETAILS

We use NVIDIA L20 GPUs with 48GB VRAM on an Ubuntu 22.04. The inference evaluation is conducted based on the *lmms-eval* (Zhang et al., 2024a) library. For LLaVA-Video, we follow its original setting of 64 input frames. To ensure a fair comparison of computational efficiency and resource usage under similar conditions, we establish a consistent setup by also setting the input frame count for NVILA to 64. As for the hyperparameters, we set the similarity threshold $\tau_{\text{sim}} = 0.8$,

| Method | VideoMME | | | | LongVideoBench | | | EgoSchema |
|--------|----------|--------|------|---------|----------|------------|---------|-----------|
| | Short | Medium | Long | Overall | Relation | Perception | Overall | |
| **Upper Bound (Full Performance)** | | | | | | | | |
| NVILA-Video | - | - | - | 61.5 | - | - | 56.3 | 52.9 |
| **Token Retain Ratio $r = 30\%$** | | | | | | | | |
| FastV | 69.4 | 55.3 | 48.8 | 57.9 | 50.8 | 55.6 | 53.0 | 49.7 |
| PruMerge | 71.0 | 53.7 | **49.9** | 58.2 | 49.3 | 58.1 | 53.4 | 47.5 |
| FasterVLM | 72.6 | 56.9 | 51.0 | 60.1 | 48.7 | 57.8 | 53.0 | 49.3 |
| VisionZip | 71.4 | 56.2 | 49.7 | 59.1 | 46.3 | 56.2 | 50.9 | 48.9 |
| FrameFusion | 72.1 | 55.7 | 48.7 | 58.8 | 51.3 | **59.3** | 54.9 | 51.3 |
| **ST-SimDiff (Ours)** | **73.3** | **57.7** | 49.7 | **60.2** | **51.7** | 59.2 | **55.2** | **51.7** |
| **Token Retain Ratio $r = 50\%$** | | | | | | | | |
| FastV | 71.9 | 58.3 | 49.3 | 58.9 | 49.5 | 58.8 | 53.9 | 50.2 |
| PruMerge | 71.2 | 54.8 | 46.8 | 57.6 | 49.7 | 58.7 | 53.9 | 48.9 |
| FasterVLM | 74.6 | 57.8 | 50.1 | 60.8 | 48.5 | 58.1 | 53.0 | 50.5 |
| VisionZip | 72.8 | 58.1 | 50.6 | 60.5 | 50.4 | 58.9 | 54.4 | 50.3 |
| FrameFusion | 72.7 | 57.2 | 48.3 | 59.4 | 51.0 | 59.2 | 54.8 | **52.6** |
| **ST-SimDiff (Ours)** | **73.9** | **59.7** | **51.4** | **61.7** | **52.3** | **61.3** | **56.5** | 52.5 |

Table 2: Performance comparison on long-form video understanding benchmarks on NVILA-Video-8B for 64 input frames (%). The best performance among all methods is emphasized in **bold**.

and the difference threshold $\tau_{\text{diff}} = 0.2$. Experiments are conducted with token compression rates of $r = 30\%$ and $r = 50\%$, respectively. In our practical implementation, considering the trade-off between time consumption and performance improvement, we opt to use connected component finding as community detection algorithm.

## 5.3 COMPARISON WITH STATE-OF-THE-ARTS

To comprehensively evaluate our proposed ST-SimDiff framework, we compare it against a range of mainstream efficient video language model (VLM) compression methods. All experiments are conducted on three widely-used long-form video understanding benchmarks: VideoMME, LongVideoBench, and EgoSchema. To validate the generalization ability and robustness of our method, we report detailed performance data on two different base models, LLaVA-Video-7B and NVILA-8B, under token retain ratios of 30% and 50%. The experimental results are presented in Table 1 and Table 2.

(1) ST-SimDiff consistently outperforms all competing methods across all test configurations. A particularly noteworthy finding is that at a 50% token retain ratio, the overall performance of our method on both base models not only surpasses other compression algorithms but, on some benchmarks, even matches or exceeds that of the original model using 100% of the tokens. This result validates the effectiveness and novelty of ST-SimDiff and prompts a deeper analysis of the strategies and limitations of existing baselines.

(2) In importance-based baseline methods (e.g., FastV, FasterVLM), FasterVLM typically exhibits relatively stronger performance. However, their common limitation lies in the inefficient handling of temporal redundancy prevalent in videos, as they tend to retain important yet repetitive tokens, thereby limiting information compression efficiency. Hybrid baseline methods (e.g., PruMerge, VisionZip, FrameFusion) demonstrate more competitive performance than importance-only approaches, with FrameFusion generally showing the most outstanding overall results. Nevertheless, the common optimization goal of these methods is still to identify and preserve representative "com-

| Method | | VideoMME | LongVideoBench | EgoSchema |
|---|---|---|---|---|
| **Token Retain Ratio $r = 30\%$** | | | | |
| Baseline | | 60.3 | 56.2 | 54.8 |
| + Sim | Spatial | 61.5 | 56.5 | 55.2 |
| | Temporal | 61.7 | 56.8 | 55.1 |
| | Spa. + Tem. | 62.6 | 57.0 | 55.3 |
| **++ Diff (Proposed)** | | **63.2** | **57.5** | **56.0** |
| **Token Retain Ratio $r = 50\%$** | | | | |
| Baseline | | 63.2 | 56.7 | 56.5 |
| + Sim | Spatial | 63.3 | 57.0 | 56.7 |
| | Temporal | 63.7 | 57.2 | 56.9 |
| | Spa. + Tem. | 63.7 | 57.8 | 57.2 |
| **++ Diff (Proposed)** | | **63.8** | **57.9** | **57.3** |

Table 3: Ablation results of different components (%).

monality" information, with a theoretical blind spot of potentially overlooking key narrative information driven by "turning points" and "abrupt events".

(3) In summary, existing baselines reflects a deepening understanding of "redundancy" in videos, yet they still face two core challenges: first, a lack of synergistic analysis of spatio-temporal joint similarity between visual tokens, making it difficult to capture complex redundancies; and second, a general neglect of "difference" detection, failing to actively preserve key changes that define plot development. The performance gain of ST-SimDiff stems from addressing these issues by constructing a spatio-temporal graph to uniformly model complex spatio-temporal relationships and innovatively introducing difference detection.

## 5.4 ABLATION STUDY

To validate the effectiveness of each core component within our ST-SimDiff framework, we conducted a series of detailed ablation studies. We started with a Baseline strategy that includes only fundamental importance-based pruning and progressively introduced our proposed similarity selection module (+Sim) and difference selection module (++Diff) to quantify their respective contributions. All experiments were conducted at token retain ratios of 30% and 50%, with the results presented in Table 5.

(1) First, we evaluated the effectiveness of the +Sim module. The experimental results clearly indicate that introducing similarity modeling on top of the baseline consistently improves model performance. We further broke down the similarity strategy into three types: spatial-only (Spatial), temporal-only (Temporal), and spatio-temporal joint (Spa. + Tem.). The data shows that collaboratively modeling the spatio-temporal associations between visual tokens is the most effective way to compress redundancy.

(2) After integrating the optimal spatio-temporal joint similarity module, we introduced the difference selection module (++Diff) to form our complete framework. The addition of this module provides a significant performance leap, particularly under the high compression ratio of 30%. This demonstrates its vital complementary role: while the aggressive +Sim module focuses on redundancy, the ++Diff module acts as a safety net to identify and preserve unique event tokens that define the video's narrative. In contrast, the module's marginal gain is smaller at a 50% retention ratio. This is because the more lenient +Sim module is more likely to have already captured these event tokens, and the overall model performance is already approaching its upper bound. This analysis confirms our core motivation: the framework's strength lies in its ability to balance similarity-based redundancy compression with the crucial preservation of difference-based events, especially in demanding high-compression scenarios.

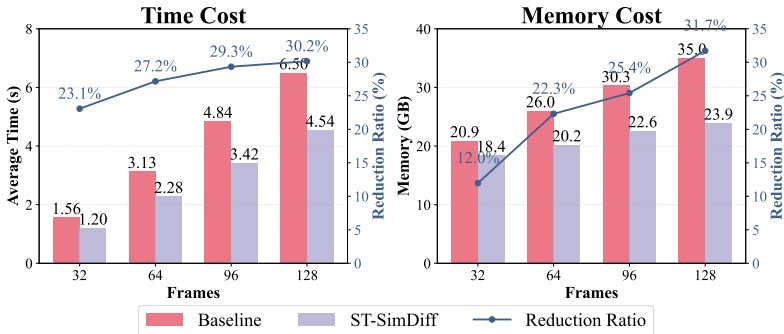

Figure 3: Computational cost comparison between our method and the baseline LLaVA-Video model in average inference time and peak GPU memory usage under a 30% token budget.

## 5.5 COMPUTATIONAL COST ANALYSIS

To quantify the efficiency advantages of our proposed ST-SimDiff method in practical applications, we conducted detailed tests on its inference time and peak GPU memory usage under a 30% token budget. We compared it against the baseline model (Baseline LLaVA-Video) without any compression. The experimental results in Figure 3 cover video inputs lengths from 32 to 128 frames.

**Inference Time**  ST-SimDiff demonstrates a significant speedup. As the number of input video frames increases, the average inference time of the baseline model grows sharply from 1.56 seconds (32 frames) to 6.50 seconds (128 frames). In contrast, our method effectively curtails this growth, with an average time of only 4.54 seconds when processing 128 frames. This means that ST-SimDiff achieves an increasingly higher time-saving rate for longer videos, improving from 23.0% for 32 frames to 30.2% for 128 frames.

**Memory Cost**  ST-SimDiff also performs exceptionally well. The peak GPU memory usage of the baseline model climbs linearly with video length, from 20.9 GB (32 frames) to 35.0 GB (128 frames). Our method significantly reduces memory pressure by substantially decreasing the number of tokens fed into the large language model. When processing 128-frame videos, ST-SimDiff's peak memory usage is only 23.9 GB, saving 31.7% of the memory space compared to the baseline. This advantage enables our model to process longer videos on hardware with limited memory, greatly expanding its potential applications.

In summary, ST-SimDiff not only achieves leading performance but also demonstrates significant advantages in computational efficiency (both time and memory), proving its efficiency for video understanding.

## 6 CONCLUSION

In this paper, we address the computational inefficiency of Multimodal Large Language Models when processing long videos, a problem rooted in the excessive number of visual tokens and the neglect of critical content changes. We propose ST-SimDiff, a framework requiring no training that builds a spatiotemporal graph to model complex associations between tokens. The method utilizes a dual selection strategy where similarity based selection identifies redundancy via community detection and temporal difference based selection preserves tokens at key transition points. Extensive experiments on VideoMME and LongVideoBench show that ST-SimDiff significantly reduces processing time and memory usage while consistently surpassing state of the art performance. This work introduces a new research direction by balancing content representativeness with transitional events to create a more efficient representation of video data.

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

# A    VISUALIZATION OF THE TOKEN SELECTION PROCESS

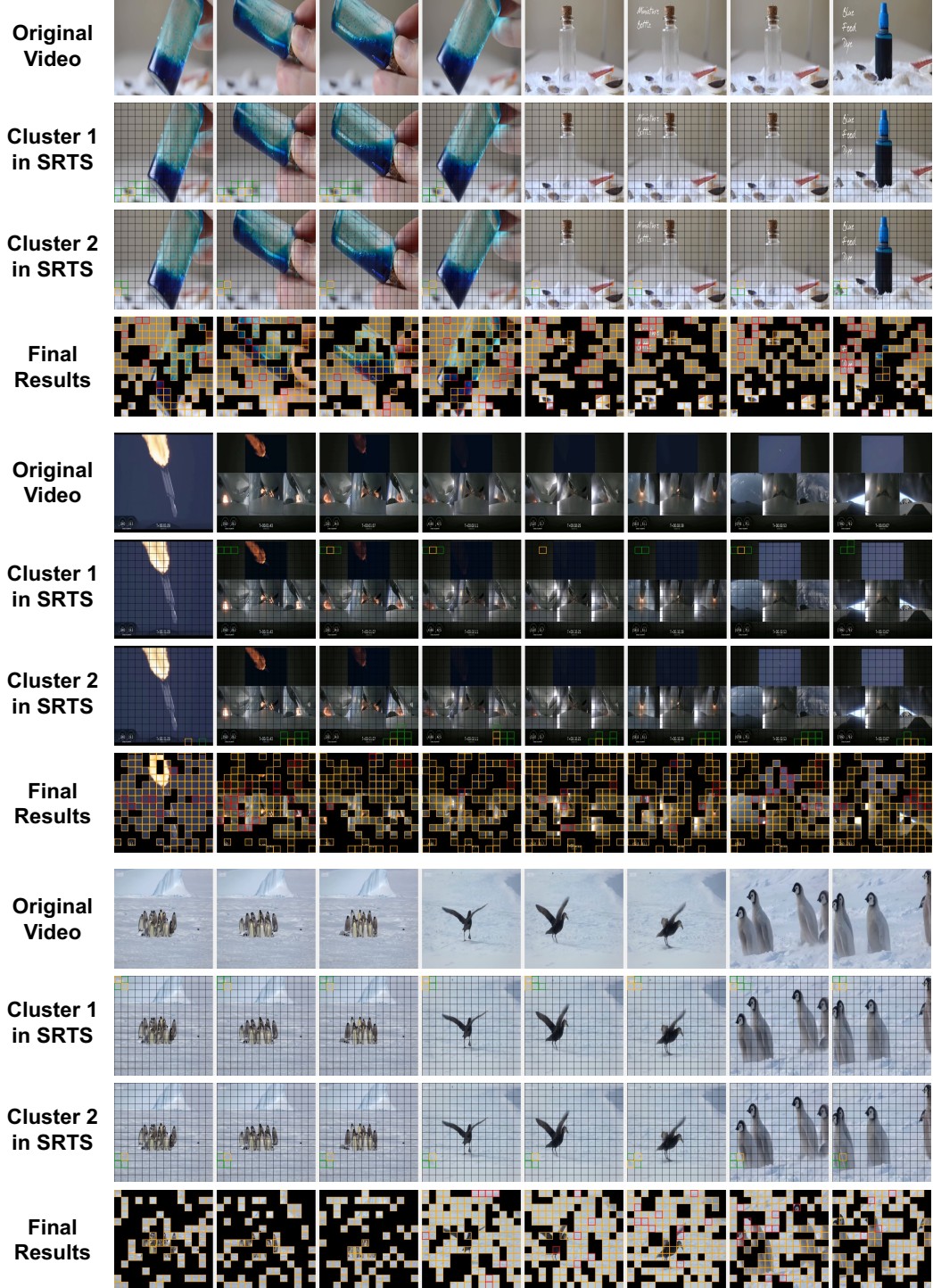

Figure 4: Visualization of the ST-SimDiff process. The visualization breaks down the token selection into communities detected by SRTS (shown as grids in Cluster 1 and 2) and event-driven tokens detected by DETS. The final result highlights the synergy between sparse representative tokens (yellow) for stable content and dense event tokens (red) for dynamic actions.

To provide a more intuitive understanding of the ST-SimDiff framework, we present some visualization samples of the intermediate and final token selection results in Figure 4. The figure illustrates the process on a sample video sequence featuring a dynamic object manipulation task. First, regarding the Similarity-based Representative Token Selection (SRTS), the rows labeled "Cluster 1" and "Cluster 2" demonstrate how our graph community detection algorithm functions. It successfully groups spatially and temporally redundant regions, such as the static background and stationary objects, into cohesive communities. This visualization confirms that SRTS effectively identifies stable visual elements and compresses them by selecting only a few central "representative tokens," which appear as sparse yellow boxes in the final results. This mechanism allows the model to handle massive background redundancy efficiently.

Complementing this, the Difference-based Event Token Selection (DETS) plays a critical role during dynamic moments. As seen in the "Final Results" row, tokens triggered by high temporal difference are explicitly marked with dense red bounding boxes. These red boxes align perfectly with the moving hand and the shifting blue liquid, verifying that our method precisely locates "turning points" where the temporal difference exceeds the threshold ($\tau_{diff}$). This ensures that fine-grained action details and rapid visual changes are preserved, allowing the model to capture the essence of the event without being overwhelmed by static data.

Finally, the synergistic effect of these two components is evident in the overlay of the final results. The visualization demonstrates a powerful balance: the sparse yellow tokens represent the stable background, while the dense red tokens capture the rapid motion. This contrast provides direct visual evidence of our core motivation. ST-SimDiff avoids the blind spots of similarity-only approaches by densely sampling during rapid changes, while simultaneously pruning the spatiotemporal redundancies that importance-based methods often miss. Consequently, our framework achieves an optimal balance between computational efficiency and the preservation of critical visual information.

## B    COMPARISON RESULTS ON QWEN2.5-VL

| Method | VideoMME | | | | LongVideoBench | | | EgoSchema |
|---|---|---|---|---|---|---|---|---|
| | Short | Medium | Long | Overall | Relation | Perception | Overall | |
| **Upper Bound (Full Performance)** | | | | | | | | |
| Qwen2.5-VL-64 | 74.2 | 62.7 | 51.8 | 62.9 | 54.2 | 64.8 | 59.2 | 63.0 |
| Qwen2.5-VL-128 | 77.4 | 65.8 | 56.1 | 66.4 | 56.2 | 65.1 | 60.4 | 63.9 |
| **Token Retain Ratio $r = 30\%$** | | | | | | | | |
| ST-SimDiff-64 | 73.8 | 61.7 | 51.8 | 62.4 | 52.2 | 62.1 | 56.8 | 62.7 |
| ST-SimDiff-128 | 75.4 | 64.7 | 55.2 | 65.1 | 53.1 | 65.0 | 58.6 | 64.4 |
| **Token Retain Ratio $r = 50\%$** | | | | | | | | |
| ST-SimDiff-64 | 74.2 | 62.1 | 52.2 | 62.9 | 54.2 | 63.2 | 58.4 | 62.7 |
| ST-SimDiff-128 | 76.8 | 66.3 | 55.8 | 66.3 | 54.9 | 65.3 | 59.8 | 64.5 |

Table 4: Performance comparison on long-form video understanding benchmarks on Qwen2.5-VL-7B for 64 and 128 input frames (%).

To further validate the generalization capability of our proposed framework, we conducted additional experiments on the recently released Qwen2.5-VL model. These evaluations were performed on the VideoMME and LongVideoBench benchmarks to confirm that our method's efficacy is not limited to a specific model architecture but can be broadly applied.

The empirical results, presented in Table 4, demonstrate the strong performance of ST-SimDiff on the Qwen2.5-VL model. At a token retention ratio of 30%, our method achieved scores of 62.4 on VideoMME and 56.8 on LongVideoBench. When the token budget was increased to a 50% retention ratio, the performance improved to 62.9 on VideoMME and 58.4 on LongVideoBench. It is particularly noteworthy that at the 50% ratio, the performance on VideoMME matches the upper-bound performance achieved using all tokens, and the score on LongVideoBench is highly comparable to its full-performance counterpart. These findings affirm the excellent generalization ability of our framework.

## C ABLATION STUDY RESULTS ON NVILA

| Method | | VideoMME | LongVideoBench | EgoSchema |
|---|---|---|---|---|
| **Token Retain Ratio $r = 30\%$** | | | | |
| Baseline | | 59.3 | 53.8 | 50.2 |
| + Sim | Spatial | 59.5 | 54.5 | 50.9 |
| | Temporal | 59.6 | 54.3 | 51.2 |
| | Spa. + Tem. | 59.9 | 54.6 | 51.3 |
| **++ Diff (Proposed)** | | **60.2** | **55.2** | **51.7** |
| **Token Retain Ratio $r = 50\%$** | | | | |
| Baseline | | 60.9 | 55.3 | 51.5 |
| + Sim | Spatial | 61.1 | 55.7 | 51.9 |
| | Temporal | 61.3 | 55.6 | 51.8 |
| | Spa. + Tem. | 61.4 | 56.0 | 52.3 |
| **++ Diff (Proposed)** | | **61.7** | **56.5** | **52.5** |

Table 5: Ablation results of different components on NVILA-Video-8B model (%).

To further validate the generalization ability of each component in our framework across different base models, we also conducted a series of detailed ablation studies on the NVILA-Video-8B model (Liu et al., 2024c). The experimental setup is consistent with the ablation study in the main paper; we start with a Baseline strategy that includes only fundamental importance-based pruning and progressively introduce the similarity selection module (+ Sim) and the difference selection module (++ Diff).

As shown in Table 5, the experimental results once again validate the effectiveness of our framework design, and the trends are highly consistent with the performance on LLaVA-Video-7B. First, after introducing the similarity module (+ Sim) on top of the baseline, the model's performance shows a steady improvement at both 30% and 50% token retain ratios. Within the similarity module, we compared three strategies: spatial-only (Spatial), temporal-only (Temporal), and spatio-temporal joint (Spa. + Tem.). The results show that spatio-temporal joint modeling (Spa. + Tem.) is the most effective. For example, with r = 50%, the spatio-temporal joint strategy achieved a score of 61.4% on VideoMME, outperforming the spatial-only and temporal-only strategies. This again demonstrates the importance of collaboratively considering spatio-temporal associations for effectively identifying redundant information. After integrating the optimal spatio-temporal joint similarity module, we further introduced the dissimilarity selection module (++ Diff) to form our complete ST-SimDiff framework. The table clearly shows that the addition of this module brought the most significant performance gains. This result strongly demonstrates the effectiveness and generalization ability of the two core modules in our framework (similarity and dissimilarity selection) and once again validates our core idea of simultaneously handling redundancy and change.

## D ABLATION RESULTS OF $\tau_{sim}$ AND $\tau_{diff}$ ON LLAVA-VIDEO

In our framework, the similarity threshold $\tau_{sim}$ and the difference threshold $\tau_{diff}$ are two key hyperparameters that respectively influence the selection of representative and transitional tokens. To investigate their impact, we conducted a series of ablation studies, with the results shown in Figure 6. The experiments show that the impact of both parameters on model performance follows a similar trend, first rising and then falling, while demonstrating good robustness within a certain range. For $\tau_{sim}$, a value that is too low leads to imprecise community detection, while a value that is too high can disrupt the integrity of semantic clusters. For $\tau_{diff}$, a value that is too low can introduce noise due to over-sensitivity, whereas a value that is too high may cause the model to miss key events. According to the experimental results, the model achieves optimal performance at $\tau_{sim}$ = 0.8 and $\tau_{diff}$ = 0.2. Therefore, we adopted this optimal configuration for all other experiments.

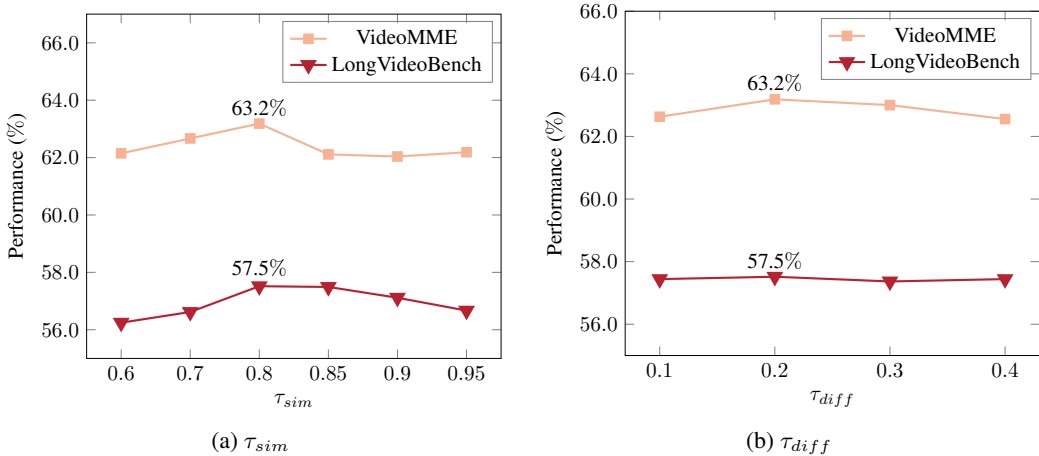

Figure 5: Ablation study results for different values of $\tau_{diff}$ and $\tau_{sim}$ on VideoMME and LongVideoBench.

# E    ABLATION RESULTS OF $\tau_{sim}$ AND $\tau_{diff}$ ON NVILA-VIDEO

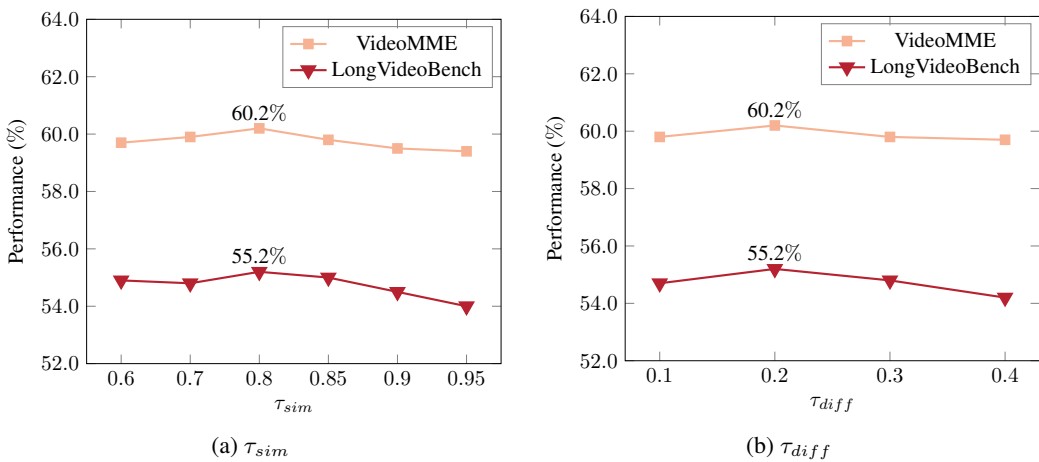

Figure 6: Ablation study results for different values of $\tau_{diff}$ and $\tau_{sim}$ on NVILA-Video-8B.

To determine the robustness of the key hyperparameters in our method, we conducted a series of ablation studies on the NVILA-Video-8B model. As shown in Fig. 6, we evaluated the impact of the similarity threshold, $\tau_{\text{sim}}$, and the difference threshold, $\tau_{\text{diff}}$, on the model's performance across two benchmarks: VideoMME and LongVideoBench.

**Impact of Similarity Threshold $\tau_{\text{sim}}$.**  We tested different values for $\tau_{\text{sim}}$ within the range of $[0.6, 0.95]$. The experimental results show that the model's performance is relatively insensitive to changes in this threshold. On the VideoMME dataset, peak performance of 60.2% was achieved at $\tau_{\text{sim}} = 0.8$. Similarly, on the LongVideoBench dataset, performance was also optimal at $\tau_{\text{sim}} = 0.8$, reaching 55.2%. While performance slightly decreased when the threshold was too high or too low, the overall fluctuation was minimal, confirming that our method maintains good stability across a wide range of similarity thresholds.

**Impact of Difference Threshold $\tau_{\text{diff}}$.**  We tested $\tau_{\text{diff}}$ in the range of $[0.1, 0.4]$. Similar to $\tau_{\text{sim}}$, the model's performance demonstrated strong robustness to variations in $\tau_{\text{diff}}$. In both benchmarks, the optimal performance was achieved around $\tau_{\text{diff}} = 0.2$, with VideoMME at 60.2% and LongVideoBench at 55.2%. This indicates that setting the difference threshold around 0.2 is most

effective for capturing key events in the video, thereby maximizing model performance while compressing tokens.

In summary, these ablation studies confirm the robustness of our proposed method with respect to its key hyperparameters. The results validate our choice of using $\tau_{\text{sim}} = 0.8$ and $\tau_{\text{diff}} = 0.2$ in our main experiments, as these values consistently yield optimal performance across different benchmarks and different base models.

