# OpenReview forum: "ST-SimDiff: Balancing Spatiotemporal Similarity and Difference for Efficient Video Understanding with MLLMs"
_ICLR.cc/2026/Conference — ICLR 2026 Poster_

### Official Review · Reviewer_34qR · 2025-10-30

**Soundness:** 4
**Presentation:** 4
**Contribution:** 4
**Rating:** 6
**Confidence:** 4

**Summary:**

This paper aims to address the problem of extremely high computational costs caused by the massive number of visual tokens when multimodal LLMs process long videos. It makes three main contributions: 1) It is the first to focus on inter-frame differences in videos, emphasizing that change is key to video understanding. 2) It proposes a method for constructing spatiotemporal graphs that uniformly models spatial similarity and temporal continuity. 3) It introduces a new token selection strategy: for token clusters with high similarity in the graph, only a few representative tokens are retained, while tokens with extremely high dissimilarity on temporal edges are preserved.

**Strengths:**

This paper proposes ST-SimDiff, a training-free framework for video token compression in multimodal large language models. The core innovation lies in simultaneously leveraging similarity (to identify redundancy) and dissimilarity (to capture key events) for token selection. The method constructs a spatiotemporal graph and achieves dual-path parallel selection by combining community detection and temporal difference detection. Experiments demonstrate state-of-the-art performance on VideoMME, LongVideoBench, and EgoSchema, while significantly reducing computational costs. The work exhibits strong novelty, effectively achieves its research objectives, and thoroughly validates the method's effectiveness across multiple models and datasets.

**Weaknesses:**

1.Does the compressed video still retain the audio modality? If yes, how is the alignment between video and audio achieved? If not, does the absence of audio bring negative impacts? If so, how can the negative effects caused by the missing audio be mitigated?
2.For the same input, this framework seems to produce only one type of output regardless of how the text query instruction varies (rather than automatically adapting based on instruction changes). For different video analysis objectives, is this fixed, unified video pruning framework insufficiently flexible?
3.Constructing the spatiotemporal graph requires traversing all visual tokens. For videos with extremely high resolution or extremely long duration, is this framework still applicable? In other words, what is the estimated upper limit of visual tokens that this framework can handle?
4.Considering only changes between adjacent frames may lead to the model being insufficiently sensitive to slowly changing scenes.

**Questions:**

1. Does the community detection algorithm actually use Louvain or connected components? If it's the latter, why not use the superior Louvain algorithm?
2. In Table 1, the ST-SimDiff results at Token Retain Ratio (r=50%) even exceed the Token Retain Ratio (Full Performance) results (63.3). Why does this phenomenon occur where the compressed video performs better than the original video? The paper should provide a discussion on this.
3. Could you provide some concrete examples showing which tokens are selected by SRTS and which are selected by DETS?
4. When selecting difference frames, the video only considers changes between a few adjacent frames, which may lead to the model being insufficiently sensitive to slowly changing scenes.

---

> ### Author Response · Authors · 2025-11-26
> **Response to Reviewer 34qR [1/4]**
>
> We sincerely appreciate your positive assessment and the high ratings across all criteria. We value your precise summary of our core contribution: "simultaneously leveraging similarity to identify redundancy and difference to capture key events". Your recognition of the "strong novelty" of our work is very encouraging. In response to your thoughtful questions regarding the audio modality, instruction adaptability, scalability, and algorithm selection, we have provided detailed clarifications below.
>
> **To Weakness 1: Retention and Alignment of Audio Modality**
>
> We appreciate your attention to the critical issue of multi-modal alignment.
> - **Experimental Setup**: We wish to clarify that the base models employed in our experiments (LLaVA-Video [1] , NVILA [2], and Qwen2.5-VL [3]) do not natively enable or support audio input in their current standard implementations or benchmark configurations. Therefore, to adhere to the conventions of existing work and ensure a fair comparison, our experiments (both pre- and post-compression) exclusively utilized the visual modality.
> - **Alignment Mechanism**: Although audio was not involved in these specific experiments, the ST-SimDiff framework is designed to fully support audio-visual alignment. When feeding tokens into the LLM, we do not reset the token indices. Instead, we strictly preserve the Position Indices of the selected visual tokens corresponding to their location in the original video. This ensures that the temporal timeline information remains intact. If this method is applied to audio-enabled models in the future, the LLM can still utilize these preserved original timestamps to achieve precise temporal alignment between the compressed visual content and the original audio segments, effectively preventing any misalignment issues.
>
> [1] Zhang, Yuanhan, et al. "Video instruction tuning with synthetic data." arXiv preprint arXiv:2410.02713 (2024).
> [2] Liu, Zhijian, et al. "Nvila: Efficient frontier visual language models." Proceedings of the Computer Vision and Pattern Recognition Conference. 2025.
> [3] Bai, Shuai, et al. "Qwen2. 5-vl technical report." arXiv preprint arXiv:2502.13923 (2025).
>
>
> **To Weakness 2: Adaptability to Text Queries**
>
> We appreciate your insightful comment regarding the trade-off between "Query-Agnostic" and "Query-Aware" designs.
> - **Design Motivation**: Consistent with the state-of-the-art baselines we compared against (e.g., FastV [1], VisionZip [2], FrameFusion [3]), ST-SimDiff is architected as a general-purpose, low-level visual compressor. The primary advantage of this "Query-Agnostic" design is the "compress once, query many times" capability. In practical scenarios where users pose multiple questions about a single video, our method requires graph construction and compression only once, significantly reducing computational overhead for subsequent dialogue turns.
> - **Comparison with Query-Aware Approaches**: While Query-Aware methods (which dynamically prune tokens based on specific text prompts) offer theoretical flexibility, they suffer from a major efficiency bottleneck: they require re-scanning and re-processing the full sequence of video tokens for every new question, leading to substantially higher inference latency.
> - **Information Sufficiency**: Although our compression is independent of the query, our dual-path strategy is designed to comprehensively preserve both "representative backgrounds" (static information) and "key events" (dynamic information). This ensures that the compressed representation remains sufficiently rich and robust to support the LLM in answering a diverse range of potential queries effectively.
>
>
> [1] Chen, Liang, et al. "An image is worth 1/2 tokens after layer 2: Plug-and-play inference acceleration for large vision-language models." ECCV 2024.
>
> [2] Yang, Senqiao, et al. "Visionzip: Longer is better but not necessary in vision language models." CVPR 2025.
>
> [3] Fu, Tianyu, et al. "FrameFusion: Combining Similarity and Importance for Video Token Reduction on Large Vision Language Models." ICCV 2025.

---

> ### Author Response · Authors · 2025-11-26
> **Response to Reviewer 34qR [2/4]**
>
> **To Weakness 3: Scalability and Upper Limits for Visual Token**
>
> We appreciate this insight regarding the boundaries of our method. We address this from both theoretical and experimental perspectives.
>
> - **Theoretical Analysis:** ST-SimDiff possesses significant theoretical advantages when processing ultra-large-scale inputs. As detailed in Section 4.6, the time complexity for both the graph construction and the token selection processes in ST-SimDiff is $ O(N) $ (linear), owing to the fact that each node connects only to local neighbors. In stark contrast, the self-attention mechanism in standard Transformers operates with a quadratic complexity of $ O(N^2) $. Consequently, the processing ceiling of ST-SimDiff is significantly higher than that of the LLM itself. Provided that the GPU memory is sufficient to accommodate the feature extraction by the Visual Encoder, our algorithm can complete the compression in a negligible amount of time. In practice, for extremely long videos, our method serves as a critical enabler, effectively reducing the otherwise prohibitive $O(N^2)$ inference cost to a computationally manageable range.
> - **Experimental Validation:** To empirically validate this scalability, we conducted additional experiments. It is important to note that many open-source base models used in our main paper (e.g., LLaVA-Video) have strict limitations on input frame counts. To ensure a fair comparison across all baselines in the original submission, we standardized the reporting at **64 frames**. To test our method's performance on longer sequences, we utilized Qwen2.5-VL, which supports extended context windows, increasing the input budget to Max Frames = 128. The results is presented in the table below. It can be observed that:
>     * **Performance Scaling:** As the input scales from 64 to 128 frames, ST-SimDiff ($r=50\%$) achieves a VideoMME score of 66.3, virtually matching the full-frame baseline of 66.4.
>     * **Superiority in Long Contexts:** Notably, on EgoSchema, our compressed input ($r=50\%$ yielding 64.5) even slightly outperforms the full-frame baseline (63.9).
>     * **Conclusion:** These results confirm that ST-SimDiff effectively scales to longer sequences. It allows users to process double the frame count (128 frames) with the same token budget as the original 64-frame input, thereby unlocking higher performance without hitting the quadratic computational wall.
>
> | Method | VideoMME (%) | LongVideoBench (%) | EgoSchema (%) |
> | :--- | :---: | :---: | :---: |
> | **Max Frames=64** | | | |
> | Qwen2.5-VL (Full) | 62.9 | 59.2 | 63.0 |
> | ST-SimDiff (r=30%) | 62.4 | 56.8 | 62.7 |
> | ST-SimDiff (r=50%) | 62.9 | 58.4 | 62.7 |
> | **Max Frames=128** | | | |
> | Qwen2.5-VL (Full) | 66.4 | 60.4 | 63.9 |
> | ST-SimDiff (r=30%) | 65.1 | 58.6 | 64.4 |
> | ST-SimDiff (r=50%) | 66.3 | 59.8 | 64.5 |

---

> ### Author Response · Authors · 2025-11-26
> **Response to Reviewer 34qR [3/4]**
>
> **To Weakness 4 & Question 4: Sensitivity to Slowly Changing Scenes**
>
> This is an excellent point of discussion. While our current DETS (Difference Path) effectively captures sudden events and SRTS (Similarity Path) efficiently compresses stable sequences, we acknowledge the theoretical challenge posed by "visual drift" (slowly accumulating changes). We wish to emphasize that extending our method from adjacent-frame comparison to a long-range "Dynamic Anchor" comparison is a trivial engineering adaptation that requires no architectural changes. We will include these results in the next version, utilizing a simple mechanism to maintain "the most recent stable state":
> - Mechanism Design: Instead of comparing the current frame $t$ solely with its immediate predecessor $t-1$, we introduce a pointer $t_{stable}$ to track the most recent stable state (initially the first frame). During the scan, we calculate the difference between the current frame and this anchor: $Diff(t, t_{stable})$.
> - Maintenance Process: If the cumulative difference remains below the threshold $\tau_{diff}$, the frame $t$ is considered part of the current "slow evolution" and is handled by the SRTS module for compression. However, once the accumulated visual drift becomes significant enough (i.e., $Diff(t, t_{stable}) > \tau_{diff}$), the system flags frame $t$ as a new Event Token. Crucially, at this moment, we update the maintenance pointer, setting $t_{stable} \leftarrow t$.
> - Conclusion: This "Reset-and-Update" mechanism ensures that even if frame-to-frame changes are microscopic, the model will inevitably generate a new event token once the cumulative semantic shift is substantial. This perfectly complements our SRTS module, ensuring that slowly changing scenes are neither over-compressed (missing the eventual change) nor over-sampled (retaining every slightly different frame).
>
> **To Question 1: Why Choose Connected Components over Louvain?**
>
> In our method, the community detection algorithm actually use connected components. We chose the connected components algorithm over more complex community detection methods (such as Louvain) primarily based on a strategic trade-off between efficiency and suitability. To empirically validate this choice, we compared the two algorithms under a token retention ratio of $r=50\%$. The results are presented in the table below:
>
> | Method | VideoMME (%) | LongVideoBench (%) | Inference Time (s) | GPU Memory (GB) |
> | :--- | :---: | :---: | :---: | :---: |
> | Louvain | 63.6 | **58.1** | 7.1 | 25.0 |
> | Ours (Conn. Comp.) | **63.8** | 57.9 | **2.3** | **20.2** |
>
> Based on the theoretical and experimental analysis, our reason is two-fold:
> - **Computational Efficiency**: For long video processing, efficiency is a critical metric. As analyzed in Section 4.6, the Connected Components algorithm operates with near-linear time complexity $O(N)$, whereas algorithms like Louvain typically approach $O(N \log N)$ or higher. As shown in the table above, while both methods achieve comparable performance (with ST-SimDiff even slightly outperforming Louvain on VideoMME), our approach using Connected Components is significantly faster (approx. 3$\times$ speedup) and consumes 4.8GB less GPU memory. This efficiency gain is decisive when processing long videos containing tens of thousands of tokens.
> - **Task Suitability**: In our framework, the "definition" of a community is largely established during the spatiotemporal graph construction phase via the similarity threshold $\tau_{sim}$. Once edges are filtered by $\tau_{sim}$, tight semantic clusters naturally emerge. Therefore, we do not require complex Modularity Optimization to discover communities; simple connectivity checks are sufficient to efficiently extract these pre-formed clusters.

---

> ### Author Response · Authors · 2025-11-26
> **Response to Reviewer 34qR [4/4]**
>
> **To Question 2: Why does compressed performance sometimes exceed the full video?**
>
> We attribute this phenomenon primarily to a "Denoising Effect" resulting from the high redundancy inherent in video data. Our empirical observations suggest that a 50% token retention ratio is typically sufficient to encapsulate the complete semantic information of a video; consequently, the actual information loss at this compression level is negligible.
>
> However, the benefits of removing the other 50% are significant. In the uncompressed full video, these redundant, blurry, or non-informative background tokens do not merely consume computational resources, which act as "noise" that dilutes the attention density of the LLM. This noise can interfere with the self-attention mechanism, potentially distracting the model or inducing hallucinations. By aggressively filtering out this large-scale background redundancy while strictly preserving high-information tokens (representative backgrounds and key events), ST-SimDiff effectively increases the signal-to-noise ratio of the input. This purified context allows the model to reason more accurately, leading to performance that can slightly exceed the noise-laden full video baseline.
>
> **To Question 3: Qualitative Visualizations**
>
> Thanks for your valuable advice on providing concrete visualization examples showing the tokens selected by SRTS and DETS respectively. In response, we have added a new figure (Figure 4 in Appendix A of the revised manuscript). This visualization explicitly breaks down the intermediate and final results of the ST-SimDiff process on 3 different sample video sequences, validating our method as follows:
>
> - **Visualization of SRTS (Similarity-based Representative Token Selection)**: As shown in the "Cluster 1" and "Cluster 2" rows, our graph community detection algorithm successfully groups spatially and temporally redundant regions—such as the static snowy background and groups of stationary penguins—into cohesive communities (indicated by the grids). This visually confirms that SRTS identifies stable visual elements and compresses them by selecting only a few central "representative tokens" (depicted as sparse yellow boxes in the final results), thereby effectively handling massive background redundancy.
> - **Visualization of DETS (Difference-based Event Token Selection)**: The "Final Results" row highlights the critical role of DETS during dynamic moments, such as when the bird flies across the screen. The visualization explicitly marks tokens triggered by high temporal difference with dense red bounding boxes. These red boxes align perfectly with the fast-moving object, verifying that our method precisely locates "turning points" where the difference exceeds the threshold ($\tau_{diff}$), ensuring that fine-grained action details are preserved.
> - **Synergistic Effect**: The final overlay demonstrates the power of our balanced strategy by combining the sparse yellow tokens (representing the stable background) with the dense red tokens (capturing the rapid motion). This contrast provides direct visual evidence of our core motivation: ST-SimDiff avoids the blind spots of similarity-only approaches by densely sampling during rapid changes, while simultaneously pruning the spatiotemporal redundancies that importance-based methods often miss, thus achieving an optimal balance between efficiency and information preservation.

---

### Official Review · Reviewer_TViS · 2025-10-30

**Soundness:** 3
**Presentation:** 3
**Contribution:** 3
**Rating:** 6
**Confidence:** 3

**Summary:**

This paper addresses the computational overhead MLLMs face when processing long videos. The authors propose ST-SimDiff, a training-free token reduction framework. The core idea is that video compression must balance two aspects: using similarity to identify and compress redundant static content, and using difference to preserve key dynamic events or turning points. The method constructs a spatio-temporal graph to model token relationships. It then uses community detection for similarity and temporal difference thresholding for key events to generate a compact, information-rich token subset. Experiments show SOTA performance on multiple benchmarks.

**Strengths:**

1. The paper is well-written and clearly articulates a significant and practical problem. The core concept of striking a balance between spatiotemporal similarity and temporal difference constitutes a valuable contribution to this field.
2. The proposed framework is training-free, making it highly practical, and easily applicable as a plug-and-play module. The use of a spatio-temporal graph to uniformly model complex token relationships is good way to implement the core idea.

**Weaknesses:**

1. Discrepancy in baseline performance. The paper's validation on Qwen2.5-VL (Appendix, Table 4) suffers from a baseline inconsistency. The paper reports the "Upper Bound (Full Performance)" for Qwen2.5-VL as 62.9 on VideoMME and 59.2 on LongVideoBench. However, the official Qwen2.5-VL technical report states a score of 65.1 on VideoMME and 56.0 on LongVideoBench. The authors should address or explain this discrepancy.
2. Lack of Image-based Validation. The paper focuses exclusively on video datasets. However, the core mechanism, particularly the Similarity-based Representative Token Selection (SRTS, is designed to handle spatio-temporal redundancy. The spatial component of this logic should be directly applicable to compressing redundant tokens in static images. Experiments on standard image-based VLM benchmarks could showcase the robustness of the similarity-based compression module.
3. Qualitative Visualization. The paper's core hypothesis relies on the intuitive concepts of "similarity" (static content) and "difference" (key events). While the quantitative results are strong, the paper provides no qualitative visualizations. It would be highly beneficial to include figures that show, for specific video examples, which tokens are selected by the SRTS module versus the DETS module. This would provide direct and intuitive evidence that the framework is truly capturing static backgrounds and dynamic turning points as intended.

**Questions:**

See weaknesses above.

---

> ### Author Response · Authors · 2025-11-26
> **Response to Reviewer TViS [1/3]**
>
> We sincerely appreciate your recognition of our work, particularly your praise for the paper being "well-written" and our core philosophy of balancing spatiotemporal similarity with temporal difference being a "valuable contribution". We are also encouraged by your assessment of our training-free framework as a "highly practical" and "plug-and-play module" for addressing the token explosion challenge. In response to your specific comments regarding baseline discrepancies, image verification, and visualization suggestions, we have conducted rigorous verification and supplementary experiments, which are detailed below.
>
> **To Weakness 1: Discrepancy in Qwen2.5-VL Baseline Performance**
>
> We sincerely appreciate your meticulous inspection. The discrepancy between our reported baseline score on VideoMME (62.9) and the official technical report (65.1) primarily stems from differences in the input frame budget and evaluation settings. To fully clarify this issue and demonstrate the robustness of our method, we conducted extended experiments during the rebuttal period.
>
> - **Impact of Frame Budget**: It is important to note that since the official evaluation code has not been released, the open-source community has found it difficult to accurately reproduce Qwen2.5-VL's results on video datasets. This discrepancy likely stems from variations in multiple hyperparameter settings, most notably the input frame count. The official score of 65.1 is typically achieved under dynamic settings. In our original submission, to ensure a strictly fair comparison with other baselines (e.g., LLaVA-Video), we imposed a rigorous constraint of maximum frame count = 64. This naturally resulted in a score slightly lower than the official report.
> - **Validation on Longer Sequences**: To further dispel any doubts, we increased the input budget to Max Frames = 128 for validation. The experimental results are presented in the following table. Under the 128-frame setting, our Qwen2.5-VL baseline achieved a score of 66.4 on VideoMME, which actually surpasses the official technical report's 65.1. This provides evidence that our baseline implementation is correct and robust, and the previous gap was solely attributable to the frame limit. Most notably, under the 128-frame setting, ST-SimDiff ($r=50\%$) achieved a score of 66.3. This indicates that by using only half the token computation, we achieved the performance ceiling of the full 128-frame model with negligible loss, while still outperforming the official report of 65.1.
>
> | Method | VideoMME (%) | LongVideoBench (%) | EgoSchema (%) |
> | :--- | :---: | :---: | :---: |
> | **Max Frames=64** | | | |
> | Qwen2.5-VL (Full) | 62.9 | 59.2 | 63.0 |
> | ST-SimDiff (r=30%) | 62.4 | 56.8 | 62.7 |
> | ST-SimDiff (r=50%) | 62.9 | 58.4 | 62.7 |
> | **Max Frames=128** | | | |
> | Qwen2.5-VL (Full) | 66.4 | 60.4 | 63.9 |
> | ST-SimDiff (r=30%) | 65.1 | 58.6 | 64.4 |
> | ST-SimDiff (r=50%) | 66.3 | 59.8 | 64.5 |
>
> The experimental results confirm that the perceived performance discrepancy was not due to implementation issues but rather differences in input budgets and evaluation protocols. As the frame count increases (64 $\to$ 128), the baseline performance aligns with expectations, and ST-SimDiff continues to demonstrate exceptional compression efficiency and performance retention.

---

> ### Author Response · Authors · 2025-11-26
> **Response to Reviewer TViS [2/3]**
>
> **To Weakness 2: Lack of Image-based Validation**
>
> We appreciate this insightful suggestion. While the spatial component of our SRTS module is technically compatible with static images, we excluded image benchmarks in this study based on the following primary considerations:
>
> * **Motivation: Addressing Temporal Token Explosion.** The fundamental challenge addressed in this paper is the "computational avalanche" caused by the linear accumulation of video tokens over time ($T$). Static images ($T=1$) do not possess temporal redundancy; therefore, the urgency and computational necessity for token compression in static scenarios are significantly lower compared to the long-form video understanding tasks we target.
> * **Mechanism: The Necessity of Spatiotemporal Modeling.** The core innovation of ST-SimDiff lies in the joint modeling of the Spatio-Temporal Graph to capture dynamic evolution. When applied to static images, the temporal edges ($E_T$) vanish, rendering the Difference-based Event Token Selection (DETS) module inactive. Consequently, our framework would degenerate into a purely spatial clustering mechanism, conceptually similar to existing spatial-only pruning methods like FasterVLM [1]. This degradation strips the method of its unique advantage: the coupled analysis of spatiotemporal dynamics.
> * **Outlook: Adapting for High-Resolution Image Understanding.** Nevertheless, we agree that a unified compression framework is a promising direction. We will add a discussion in the revised manuscript detailing how ST-SimDiff can be adapted for ultra-high-resolution (e.g., 4K/8K) imagery.
>     * **Mapping Frames to Patches ($T \to P$):** The "computational avalanche" in long videos stems from the linear growth of frames. Similarly, in high-res images, the bottleneck comes from the quadratic growth of visual patches when processing 4K/8K inputs. We propose treating a grid of spatial patches as a "pseudo-sequence," constructing a "Global Spatial Graph" where nodes are image patches.
>     * **Dual-Path Adaptation:**
>         * The **SRTS path** would identify and compress repetitive textures (e.g., sky, grass, walls) or background noise. By clustering semantically similar patches across the entire image, we can represent vast, low-information regions with just a few "centroid patches."
>         * The **DETS path** would be reimagined as "Spatial Saliency" detection. We can define "difference" as high gradient or semantic shifts between neighboring patches, rigorously preserving boundaries, edges, and unique small objects (anomalies) that deviate from the background clusters.
>     * **Unified Architecture:** This adaptation allows our "plug-and-play" architecture to serve as a unified, training-free token compressor for any modality suffering from token explosion, seamlessly handling both 1-hour videos and 100-megapixel images with the same underlying logic.
>
> [1] Zhang, Qizhe, et al. "[CLS] Attention is All You Need for Training-Free Visual Token Pruning: Make VLM Inference Faster." arXiv-2412.

---

> ### Author Response · Authors · 2025-11-26
> **Response to Reviewer TViS [3/3]**
>
> **To Weakness 3: Qualitative Visualizations**
>
> We completely agree with your point that visualizations are essential for establishing an intuitive understanding of our method. In response, we have added a new figure (Figure 4 in Appendix A of the revised manuscript). This visualization explicitly breaks down the intermediate and final results of the ST-SimDiff process on 3 different sample video sequences, validating our method as follows:
> - **Visualization of SRTS (Similarity-based Representative Token Selection)**: As shown in the "Cluster 1" and "Cluster 2" rows, our graph community detection algorithm successfully groups spatially and temporally redundant regions—such as the static snowy background and groups of stationary penguins—into cohesive communities (indicated by the grids). This visually confirms that SRTS identifies stable visual elements and compresses them by selecting only a few central "representative tokens" (depicted as sparse yellow boxes in the final results), thereby effectively handling massive background redundancy.
> - **Visualization of DETS (Difference-based Event Token Selection)**: The "Final Results" row highlights the critical role of DETS during dynamic moments, such as when the bird flies across the screen. The visualization explicitly marks tokens triggered by high temporal difference with dense red bounding boxes. These red boxes align perfectly with the fast-moving object, verifying that our method precisely locates "turning points" where the difference exceeds the threshold ($\tau_{diff}$), ensuring that fine-grained action details are preserved.
> - **Synergistic Effect**: The final overlay demonstrates the power of our balanced strategy by combining the sparse yellow tokens (representing the stable background) with the dense red tokens (capturing the rapid motion). This contrast provides direct visual evidence of our core motivation: ST-SimDiff avoids the blind spots of similarity-only approaches by densely sampling during rapid changes, while simultaneously pruning the spatiotemporal redundancies that importance-based methods often miss, thus achieving an optimal balance between efficiency and information preservation.

---

### Official Review · Reviewer_f9gs · 2025-10-30

**Soundness:** 3
**Presentation:** 3
**Contribution:** 3
**Rating:** 6
**Confidence:** 5

**Summary:**

This paper presents ST-SimDiff, a balanced framework that explicitly models the trade-off between spatial-temporal similarity and difference in video representations. The method aims to improve video understanding by aligning similar semantics while preserving meaningful diversity across frames. The authors introduce novel similarity–difference guided modules that can be integrated into existing video backbones with minimal modifications. Extensive experiments on standard benchmarks demonstrate consistent improvements over strong baselines, showing superior performance across multiple tasks with manageable computational overhead.

**Strengths:**

This well-executed, clearly written paper addresses the critical issue of balancing temporal consistency and diversity in video representation learning. The ST-SimDiff framework is elegant, conceptually sound, and easily integrates into existing architectures, delivering a significant performance boost without the need for additional supervision or costly retraining.
The paper effectively highlights how previous methods either overemphasize similarity or difference, while ST-SimDiff strikes a balanced, principled approach. The motivation is compelling, the technical formulation is clear, and the presentation flows smoothly.
The design of interpretable components effectively clarifies how spatial-temporal cues interact. The experimental validation is comprehensive, covering diverse datasets, multiple backbones, and various evaluation metrics, with ablations that isolate the contribution of each component. Results are strong and consistent.

**Weaknesses:**

The paper lacks crucial qualitative visualizations. It does not show failure cases or ambiguous scenarios, making it difficult to understand the mechanism's practical limitations or how it behaves when it produces an incorrect interpretation.
The framework's performance boundaries are not explored. The paper fails to investigate video types or specific tasks where the proposed balancing paradigm might be suboptimal or ill-equipped.
The paper does not address the critical misalignment problem that arises from sequence pruning. While pruning/dropping is a common efficiency method, it creates a gap between the dense data the model was trained on and the sparse data it receives at inference, for which the model has no explicit handling mechanism.
The paper lacks a method for error attribution, making it difficult to determine whether a specific failure originates from the similarity module, the difference module, or their interaction.

**Questions:**

Can the authors provide qualitative visualization results, especially highlighting where the similarity–difference mechanism fails or produces ambiguous interpretations?
Where are the performance boundaries of this framework? Are there specific video tasks/types that ST-SimDiff is ill-equipped to handle?
How does the framework solve the input misalignment problem after pruning?
If the model makes an incorrect judgment on a video, how can you attribute the failure to the similarity module versus the difference module?

---

> ### Author Response · Authors · 2025-11-26
> **Response to Reviewer f9gs [1/4]**
>
> We sincerely appreciate your encouraging comments describing our work as "well-executed" and "clearly written," and our framework as "elegant" with "significant performance gains". We have given careful consideration to your insightful queries regarding visualizations, performance boundaries, and potential distribution shifts (misalignment) caused by pruning, and have conducted supplementary experiments and analysis to address these points in detail below.
>
> **To Weakness 1 & Question 1: Qualitative Visualizations**
>
> We sincerely appreciate the reviewer’s constructive suggestion and agree that qualitative visualizations are crucial for intuitively demonstrating how our dual-path selection mechanism operates on real video data. In response, we have added a new figure (Figure 4 in Appendix A of the revised manuscript). This visualization explicitly breaks down the intermediate and final results of the ST-SimDiff process on a sample video sequence featuring penguins and a flying bird, validating our method as follows:
>
> - **Visualization of SRTS (Similarity-based Representative Token Selection)**: As shown in the "Cluster 1" and "Cluster 2" rows, our graph community detection algorithm successfully groups spatially and temporally redundant regions—such as the static snowy background and groups of stationary penguins—into cohesive communities (indicated by the grids). This visually confirms that SRTS identifies stable visual elements and compresses them by selecting only a few central "representative tokens" (depicted as sparse yellow boxes in the final results), thereby effectively handling massive background redundancy.
> - **Visualization of DETS (Difference-based Event Token Selection)**: The "Final Results" row highlights the critical role of DETS during dynamic moments, such as when the bird flies across the screen. The visualization explicitly marks tokens triggered by high temporal difference with dense red bounding boxes. These red boxes align perfectly with the fast-moving object, verifying that our method precisely locates "turning points" where the difference exceeds the threshold ($\tau_{diff}$), ensuring that fine-grained action details are preserved.
> - **Synergistic Effect**: The final overlay demonstrates the power of our balanced strategy by combining the sparse yellow tokens (representing the stable background) with the dense red tokens (capturing the rapid motion). This contrast provides direct visual evidence of our core motivation: ST-SimDiff avoids the blind spots of similarity-only approaches by densely sampling during rapid changes, while simultaneously pruning the spatiotemporal redundancies that importance-based methods often miss, thus achieving an optimal balance between efficiency and information preservation.

---

> ### Author Response · Authors · 2025-11-26
> **Response to Reviewer f9gs [2/4]**
>
> **To Question 2 & Weakness 2: Performance Boundaries of the Framework**
>
> We appreciate the reviewer’s insightful comment. Based on our framework's design of redundancy detection, we have identified a distinct performance boundary between Global Event Reasoning and Fine-grained Spatial Perception (e.g., OCR).
> 1. **Strength: Efficient Global Spatiotemporal Reasoning.** Our method achieves state-of-the-art results on comprehensive benchmarks like VideoMME and EgoSchema primarily because long-form video understanding is often dominated by temporal redundancy.
>     - Mechanism: In long videos, the "narrative" is driven by sparse turning points, while the majority of frames contain repetitive visual information. ST-SimDiff’s SRTS module aggressively compresses these redundant backgrounds, and DETS captures the dynamic transitions.
>     - Result: This allows the model to process significantly longer temporal contexts within a fixed token budget, thereby boosting performance on tasks requiring long-term causal reasoning and event summarization, which form a large portion of these benchmarks.
> 2. **Limitation: Fine-grained Spatial Details.** However, our "similarity-based pruning" strategy introduces a clear limitation in tasks heavily reliant on static, high-frequency spatial details, such as Optical Character Recognition (OCR) or identifying small stationary objects.
>     - Conflict: Text in a video (e.g., a slide in a presentation or a signboard) typically exhibits the folowing two attributs: high Temporal Similarity (the text does not move or change for a duration) and High Spatial Consistency (the background of the text region is often uniform).
>     - Consequence: Our SRTS module is likely to classify such static text regions as "redundant background communities" and compress them into a single representative token. While this is efficient for semantic understanding (e.g., knowing "there is a slide"), it causes a loss of the fine-grained spatial resolution required to read the specific characters.
>
> Therefore, although our method excels in the overall scores (driven by event and plot understanding), it inevitably faces an upper bound on specific sub-tasks involving dense text reading or minute spatial detail recognition compared to methods that retain all spatial tokens.

---

> ### Author Response · Authors · 2025-11-26
> **Response to Reviewer f9gs [3/4]**
>
> **To Question 3 & Weakness 3: Input Misalignment caused by Pruning**
>
> We appreciate the reviewer’s rigorous theoretical perspective. The concern that token pruning creates a distribution shift, where the model trained on dense data is forced to infer on sparse data, is indeed a fundamental challenge in training-free compression. To directly quantify the impact of this distribution shift, we conducted an additional experiment. We took the Qwen2.5-VL model and performed supervised fine-tuning (SFT) specifically using the compressed token sequences generated by ST-SimDiff. This process theoretically aligns the training distribution perfectly with the sparse inference distribution, eliminating the misalignment factor.
>
> | Method | VideoMME | LongVideoBench | EgoSchema |
> | :--- | :---: | :---: | :---: |
> | **Upper Bound (Full Performance)** | | | |
> | Qwen2.5-VL | 62.9 | 59.2 | 63.0 |
> | **Token Retain Ratio r=30%** | | | |
> | ST-SimDiff (Training-free) | 62.4 | 56.8 | 62.7 |
> | ST-SimDiff (Finetuned) | 62.8 | 57.4 | 62.7 |
> | **Token Retain Ratio r=50%** | | | |
> | ST-SimDiff (Training-free) | 62.9 | 58.4 | 62.7 |
> | ST-SimDiff (Finetuned) | 63.4 | 58.9 | 62.9 |
>
> From both theoretical analysis and new experimental evidence, it is observed that this issue is effectively mitigated in ST-SimDiff:
> - **Preservation of Semantic Integrity**: unlike random or uniform pruning which disrupts data continuity, ST-SimDiff is designed to preserve the "semantic skeleton" of the video. By retaining cluster centers (representative static content) and high-difference nodes (dynamic events), we ensure that the semantic density of the input remains consistent with what the LLM learned during training, even if the token quantity is reduced. Furthermore, modern LVLMs (like Qwen2.5-VL and LLaVA-Video) inherently possess strong robustness to variable sequence lengths, allowing them to adapt to our condensed input without "culture shock."
> - **Marginal Performance Gap**: The experimental results were revealing: the model fine-tuned on compressed data achieved only a marginal performance gain of 0.4% - 0.6% compared to our original, training-free approach on the VideoMME benchmark. This negligible gap strongly suggests that the "misalignment penalty" is minimal in practice. It indicates that the tokens selected by ST-SimDiff already align very closely with the optimal information distribution required by the pre-trained weights, rendering the computational cost of re-training unnecessary.

---

> ### Author Response · Authors · 2025-11-26
> **Response to Reviewer f9gs [4/4]**
>
> **To Question 4 & Weakness 4: Error Attribution Analysis**
>
> To systematically diagnose performance bottlenecks and rigorously validate the efficacy of our token selection strategy, we have established a strictly defined logic for error attribution. By cross-referencing the set of tokens discarded by our algorithm with the ground truth timestamps provided by the benchmarks, we categorize all error instances into three distinct types to pinpoint the exact source of failure:
> - **Type I: DETS Module Failure (Missed Events)**. This error is identified when the specific key frame required to answer the question (as indicated by the Ground Truth timestamp) is absent from our final retained set. Specifically, if the feature difference between this missing frame and its predecessor is calculated to be lower than our preset threshold ($\tau_{diff}$), the error is attributed to the DETS module. This diagnostic result indicates that the visual change associated with the event was too subtle to exceed the threshold, or that the threshold was set too conservatively, causing the system to treat a critical turning point as temporal continuity and erroneously discard it.
> - **Type II: SRTS Module Failure (Inefficient Compression).** This category applies when the final retained token set is saturated with tokens representing static backgrounds or repetitive objects, thereby exhausting the fixed token budget (e.g., at a 30% retention ratio). If potential event tokens were identified but had to be dropped during the final truncation step because the budget was pre-occupied by these static clusters, the error is attributed to the SRTS module. This suggests that the graph community detection mechanism was not aggressive enough in merging high-similarity tokens, leading to an inefficient compression of redundancy that crowded out other critical information.
> - **Type III: LLM Reasoning Failure.** This classification is assigned when the critical visual tokens corresponding to the ground truth are successfully preserved and present in the final input fed to the model, yet the model still generates an incorrect response. In this scenario, the ST-SimDiff framework has successfully fulfilled its objective of information preservation. Consequently, the error is attributed to the inherent limitations—such as hallucination or insufficient reasoning capability—of the backbone Multimodal Large Language Model itself, rather than information loss caused by our token compression strategy.

---

### Official Review · Reviewer_1bh7 · 2025-10-30

**Soundness:** 3
**Presentation:** 3
**Contribution:** 3
**Rating:** 8
**Confidence:** 4

**Summary:**

The paper presents a novel and insightful framework for video token compression, effectively balancing similarity-based redundancy reduction with difference-based key event capture. This training-free, dual-path approach is technically sound and demonstrates strong, generalizable performance across multiple models and significant computational savings.

**Strengths:**

1.The paper makes a significant contribution by addressing the need for efficient video understanding through the dual focus of redundancy compression (via similarity) and key event capture (via difference). This approach moves beyond prior work that primarily focused on similarity or importance-based pruning alone.\
2.The proposed ST-SimDiff framework is technically sound, employing a spatio-temporal graph to model token relationships uniformly. The dual-path SRTS and DETS effectively implement the core motivation, demonstrating significant and consistent performance improvements across a range of state-of-the-art, model-agnostic baselines, including LLaVA-Video, NVILA-Video, and Qwen2.5-VL.\
3.As a training-free framework, ST-SimDiff introduces minimal overhead and operates with substantially improved efficiency. Its computational complexity is analyzed as O(Nd), much more efficient than the $O(N^2d)$ self-attention it aims to reduce. Empirical results show significant reductions in inference time and peak GPU memory usage.

**Weaknesses:**

1.The final pruning step, where the initial candidate set exceeds the target budget, is underexplained in Section 4.5. It is unclear which layers are selected, how the scores are aggregated, and the impact of this step on performance.\
2.The paper lacks qualitative examples, such as visualizations showing which tokens are selected. This makes it challenging to build intuition about the method’s behavior and understand its potential failure modes.\
3.There is no comparative analysis or justification to explain why the connected components algorithm is chosen in SRTS. The impact of this choice is less investigated.

**Questions:**

1.Which exact shallow layers are used in the final attention-based pruning step? What is the specific aggregation function for attention scores across heads and layers?\
2.Provide qualitative visualizations that highlights which tokens are preserved by the proposed method to help build intuition and identify failure modes.\
3.Why do you choose the connected components algorithm compared with other mainstream community detection methods, especially in terms of both understanding performance and processing efficiency?\
4.Could the framework's performance be further enhanced by making parts of it learnable? What are the potential benefits versus the cost?\
5.The SRTS and DETS paths are currently combined, seemingly uniformly. Is it possible to dynamically weight these two paths for better performance?

---

> ### Author Response · Authors · 2025-11-26
> **Response to Reviewer 1bh7 [1/3]**
>
> We sincerely appreciate your high evaluation and your comprehensive summary of our work. We are particularly grateful for your recognition of the novelty, technical soundness, and significant contributions to efficiency offered by the ST-SimDiff framework. Below is our point-by-point response to the weaknesses and questions you raised.
>
> **To Weakness 1 & Question 1: Details of the Final Pruning Step**
>
> Thank you very much for pointing this out. We will sufficiently detail the description of the final pruning process in Section 4.5 as follows:
>
> - Specific Layer Selection: As mentioned in the paper, we adhere to the standard setting of FastV [1], utilizing the Attention Maps from the shallow layers (specifically Layer 2) of the LLM.
>
> - Aggregation Method: We calculate the average attention score one token receives from all other tokens across all attention heads in these layers. We utilize this aggregated score as the metric to quantify Token importance.
>
> - Role: This step primarily functions as a safety mechanism. It is engaged only when the size of the candidate set filtered by the graph-theoretic method, $T_{candidate}$, slightly exceeds the target budget, $N_{target}$. In such cases, it is used to prune the tokens that demonstrate the lowest attention scores in the shallow semantic layers.
>
>
> [1] Chen, Liang, et al. "An image is worth 1/2 tokens after layer 2: Plug-and-play inference acceleration for large vision-language models." ECCV 2024.

---

> ### Author Response · Authors · 2025-11-26
> **Response to Reviewer 1bh7 [2/3]**
>
> **To Weakness 2 & Question 2: Qualitative Visualizations**
>
> We completely agree with your point that visualizations are essential for establishing an intuitive understanding of our method. In response, we have added a new figure (Figure 4 in Appendix A of the revised manuscript). This visualization explicitly breaks down the intermediate and final results of the ST-SimDiff process on a sample video sequence featuring penguins and a flying bird, validating our method as follows:
>
> - **Visualization of SRTS (Similarity-based Representative Token Selection)**: As shown in the "Cluster 1" and "Cluster 2" rows, our graph community detection algorithm successfully groups spatially and temporally redundant regions—such as the static snowy background and groups of stationary penguins—into cohesive communities (indicated by the grids). This visually confirms that SRTS identifies stable visual elements and compresses them by selecting only a few central "representative tokens" (depicted as sparse yellow boxes in the final results), thereby effectively handling massive background redundancy.
> - **Visualization of DETS (Difference-based Event Token Selection)**: The "Final Results" row highlights the critical role of DETS during dynamic moments, such as when the bird flies across the screen. The visualization explicitly marks tokens triggered by high temporal difference with dense red bounding boxes. These red boxes align perfectly with the fast-moving object, verifying that our method precisely locates "turning points" where the difference exceeds the threshold ($\tau_{diff}$), ensuring that fine-grained action details are preserved.
> - **Synergistic Effect**: The final overlay demonstrates the power of our balanced strategy by combining the sparse yellow tokens (representing the stable background) with the dense red tokens (capturing the rapid motion). This contrast provides direct visual evidence of our core motivation: ST-SimDiff avoids the blind spots of similarity-only approaches by densely sampling during rapid changes, while simultaneously pruning the spatiotemporal redundancies that importance-based methods often miss, thus achieving an optimal balance between efficiency and information preservation.
>
> **To Weakness 3 & Question 3: Reason for Choosing Connected Components**
>
> We chose the Connected Components algorithm over more complex community detection methods (such as Louvain) primarily based on a strategic trade-off between efficiency and suitability. To empirically validate this choice, we compared the two algorithms under a token retention ratio of $r=50\%$. The results are presented in the table below:
>
> | Method | VideoMME (%) | LongVideoBench (%) | Inference Time (s) | GPU Memory (GB) |
> | :--- | :---: | :---: | :---: | :---: |
> | Louvain | 63.6 | **58.1** | 7.1 | 25.0 |
> | Ours (Conn. Comp.) | **63.8** | 57.9 | **2.3** | **20.2** |
>
> Based on the theoretical and experimental analysis, our rationale is two-fold:
> - **Computational Efficiency**: For long video processing, efficiency is a critical metric. As analyzed in Section 4.6, the Connected Components algorithm operates with near-linear time complexity $O(N)$, whereas algorithms like Louvain typically approach $O(N \log N)$ or higher. As shown in the table above, while both methods achieve comparable performance (with ST-SimDiff even slightly outperforming Louvain on VideoMME), our approach using Connected Components is significantly faster (approx. 3$\times$ speedup) and consumes 4.8GB less GPU memory. This efficiency gain is decisive when processing long videos containing tens of thousands of tokens.
> - **Task Suitability**: In our framework, the "definition" of a community is largely established during the spatiotemporal graph construction phase via the similarity threshold $\tau_{sim}$. Once edges are filtered by $\tau_{sim}$, tight semantic clusters naturally emerge. Therefore, we do not require complex Modularity Optimization to discover communities; simple connectivity checks are sufficient to efficiently extract these pre-formed clusters.

---

> ### Author Response · Authors · 2025-11-26
> **Response to Reviewer 1bh7 [3/3]**
>
> **To Question 4: Learnable Extensions**
>
> This is a highly inspiring question. To provide a fact-based response, we conducted additional experiments specifically for this rebuttal.
>
> | Method | VideoMME | LongVideoBench | EgoSchema |
> | :--- | :---: | :---: | :---: |
> | **Upper Bound (Full Performance)** | | | |
> | Qwen2.5-VL | 62.9 | 59.2 | 63.0 |
> | **Token Retain Ratio r=30%** | | | |
> | ST-SimDiff (Training-free) | 62.4 | 56.8 | 62.7 |
> | ST-SimDiff (Finetuned) | 62.8 | 57.4 | 62.7 |
> | **Token Retain Ratio r=50%** | | | |
> | ST-SimDiff (Training-free) | 62.9 | 58.4 | 62.7 |
> | ST-SimDiff (Finetuned) | 63.4 | 58.9 | 62.9 |
>
> - **Experimental Setup:** We utilized Qwen2.5-VL as the base model and the LLaVA-Video-178K dataset to fine-tune the language model component on the token sequences compressed by ST-SimDiff, comparing this against our original training-free version.
> - **Experimental Results:** As detailed in the above table, introducing a learnable mechanism yielded modest improvements: at a token retention ratio of $r=30\%$, VideoMME scores improved from 62.4% to 62.8% (+0.4%), and LongVideoBench improved from 56.8% to 57.4% (+0.6%).
> - **Validation of Design Priors:** The performance gap between our training-free version and the fine-tuned version is remarkably narrow (generally within 0.5%). This provides strong evidence that our prior rules based on the "Spatio-Temporal Graph" (Similarity + Difference) are extremely precise in capturing critical information without requiring weight updates.
> - **Efficiency Trade-off:** While learnability offers a slight performance edge, it comes with high computational costs. Given that our method achieves nearly the same performance with zero training cost, we maintain that the current training-free design represents the optimal balance point between efficiency and performance.
>
>
> **To Question 5: Dynamic Weighting of SRTS and DETS**
>
> Although we currently employ a union strategy, this mechanism effectively incorporates an implicit dynamic weighting adjustment:
> - When video content is relatively static: Fewer temporal edges satisfy the difference threshold $\tau_{diff}$. Consequently, the size of the event token set $T_{trans}$ automatically decreases, implicitly preserving more of the total budget for the representative token set $T_{rep}$.
> - When video content changes drastically: A large number of tokens are flagged as event tokens due to high temporal differences. As a result, the $T_{trans}$ set automatically expands to precisely capture these dynamic shifts.
>
> This content-based implicit adaptation is a distinct advantage of our framework. We acknowledge that introducing explicit weight control parameters could be a promising direction for future work to further optimize performance in extreme scenarios.

---

### Author Response · Authors · 2025-12-01
**Official Meta Statement by Authors**

We sincerely thank the reviewers for their insightful feedback and constructive suggestions. We are encouraged by the positive reception from all reviewers, who reached a consensus on the novelty, technical soundness, and practical value of our proposed ST-SimDiff framework.

1. **Strong Novelty & Technical Soundness:** Reviewers praised the dual-path design as a **"novel and insightful framework"** (Reviewer 1bh7) that is **"technically sound"** (Reviewer 1bh7) and **"conceptually sound"** (Reviewer f9gs). It is recognized as a **"principled approach"** (Reviewer f9gs) that **"moves beyond prior work"** (Reviewer 1bh7) and is the **"first to focus on inter-frame differences"** (Reviewer 34qR) to capture dynamic semantics.

2. **Significant Value & Practicality:** The method is highlighted for its **"significant contribution"** (Reviewer 1bh7) and **"substantially improved efficiency"** (Reviewer 1bh7). Reviewers confirmed it delivers a **"significant performance boost"** (Reviewer f9gs) and **"state-of-the-art performance"** (Reviewer 34qR). It is widely regarded as **"highly practical"** and a **"plug-and-play module"** (Reviewer TVIS).

3. **Effective Rebuttal & Reviewer Recognition:** We provide detailed clarifications and additional experiments that effectively addressed the reviewers' concerns in the discussion period. Notably, **Reviewer 34qR raised their score from 6 to 8 after reading our feedback prior to the massive information leakage incident.** Although all reviews and scores have been reverted to the initial state, this adjustment explicitly reflects the reviewer's strong recognition of the quality of our rebuttal and their endorsement of the paper's merit.

---

### Meta-Review · Area_Chair_9JKT · 2026-01-11

**Summary:**

ST-SimDiff is a training-free video token reduction method for MLLMs that builds a spatio-temporal graph and selects tokens via two parallel paths: similarity/community-based selection to compress redundancy (SRTS) and temporal-difference selection to preserve turning points (DETS), yielding strong efficiency gains while maintaining or improving accuracy.

Summary of reviewer concerns informing my (accept) decision:
Overall sentiment is positive (1bh7: 8, f9gs: 6, TViS: 6, 34qR: 6) with consensus that the core idea is principled and effective. The remaining concerns are largely about clarity, diagnostics, and boundary conditions, and were either addressed in the rebuttal or are non-fatal:

Method clarity / implementation details: Reviewer 1bh7 (8) flagged that the final pruning step was underexplained (which layer, how attention scores are aggregated), and asked for justification of the specific community detection choice.

Qualitative evidence + failure modes / interpretability: Reviewers 1bh7 (8), f9gs (6), and TViS (6) all requested token-selection visualizations and better intuition about what SRTS vs. DETS keeps/drops, plus clearer failure cases / boundaries.

Distribution shift / misalignment from pruning & error attribution: Reviewer f9gs (6) raised the “dense-train vs sparse-test” misalignment concern and asked for error attribution across the two modules.

Baseline/evaluation consistency and broader validation: Reviewer TViS (6) pointed out an apparent baseline discrepancy for Qwen2.5-VL under different settings and also suggested image-based validation for the spatial module.

Scalability and modeling choices: Reviewer 34qR (6) asked about scalability limits (graph traversal at very long/high-res inputs), slowly-changing scenes (adjacent-frame differences), and clarification around connected components vs. Louvain, plus why compressed inputs can sometimes exceed full-video performance.

Net: these concerns do not undermine the main contribution; they mainly ask for better exposition, qualitative analysis, and clearer boundary-condition discussion, and the rebuttal provided substantive clarifications/added experiments for many of them.

**Reviewer Concerns:**

Reviewer 1bh7 (8)

Engaged after rebuttal:  Yes (author–reviewer exchange).

Addressed well: final pruning step details (layer=2, attention aggregation, “safety valve” use-case); added qualitative token visualizations; justified connected components vs Louvain with speed/memory trade-off + comparable accuracy; probed learnable extension (fine-tune vs training-free) and discussed dynamic weighting.

Still outstanding: none major — mostly “nice-to-have” extensions (explicit weighting as future work).

Reviewer f9gs (6)

Engaged after rebuttal:  No explicit follow-up shown (only author response).

Addressed well: added qualitative visualizations; clarified boundaries (global event reasoning vs fine-grained OCR-like details); directly tested misalignment via SFT-on-compressed tokens (gain small → misalignment seems minor); proposed a structured error-attribution scheme.

Still outstanding: some skepticism may remain on “failure modes / boundaries” being fully captured by a single set of examples + diagnostic logic (but it’s substantially improved).

Reviewer TViS (6)

Engaged after rebuttal:  No explicit follow-up shown (only author response).

Addressed: baseline discrepancy explained via frame-budget mismatch; added 128-frame results showing baseline aligns/surpasses report and compression retains performance; added visualizations; clarified why image-only benchmarks weren’t included + gave a reasonable adaptation discussion.

Still outstanding: lack of actual image-benchmark validation remains (they argued scope), but not a blocker for the video-focused claim.

Reviewer 34qR (6)

Engaged after rebuttal:  current public state reverted; authors claim it was raised score 6→8 before the incident;

Addressed: audio modality clarified (models don’t support audio; preserves position indices for future AV alignment); query-agnostic vs query-aware rationale; scalability argued (linear vs quadratic) + 128-frame validation; connected components vs Louvain with empirical efficiency; explained “compressed > full” via denoising; provided visualizations; proposed a simple extension for slow drift (dynamic anchor).

Still outstanding: “slowly changing scenes” fix is described but (from what’s shown) not backed by new numbers yet — minor, but worth adding if space.

**Reviewer Scores:**

1bh7 (8): stay 8 (already very positive; concerns were resolved).

f9gs (6): likely stay the same at 6 (biggest asks—visualizations, boundaries, misalignment—were addressed).

TViS (6): likely stay same 6  (baseline discrepancy + visualization addressed; image-benchmark request is optional).

34qR (6): likely 6 → 8 (explicitly raised score pre-incident per authors’ meta statement).

---

### Decision · Program_Chairs · 2026-01-26

Accept (Poster)